# College student Fear of Missing Out (FoMO) and maladaptive behavior: Traditional statistical modeling and predictive analysis using machine learning

**Paul C. McKee**[1]*, **Christopher J. Budnick**[1], **Kenneth S. Walters**[1], **Imad Antonios**[2]

**1** Department of Psychology, Southern Connecticut State University, New Haven, CT, United States of America, **2** Department of Computer Science, Southern Connecticut State University, New Haven, CT, United States of America

* paulcmc240@gmail.com

## Abstract

This paper reports a two-part study examining the relationship between fear of missing out (FoMO) and maladaptive behaviors in college students. This project used a cross-sectional study to examine whether college student FoMO predicts maladaptive behaviors across a range of domains (e.g., alcohol and drug use, academic misconduct, illegal behavior). Participants (N = 472) completed hard copy questionnaire packets assessing trait FoMO levels and questions pertaining to unethical and illegal behavior while in college. Part 1 utilized traditional statistical analyses (i.e., hierarchical regression modeling) to identify any relationships between FoMO, demographic variables (socioeconomic status, living situation, and gender) and the behavioral outcomes of interest. Part 2 looked to quantify the predictive power of FoMO, and demographic variables used in Part 1 through the convergent approach of supervised machine learning. Results from Part 1 indicate that college student FoMO is indeed related to many diverse maladaptive behaviors spanning the legal and illegal spectrum. Part 2, using various techniques such as recursive feature elimination (RFE) and principal component analysis (PCA) and models such as logistic regression, random forest, and Support Vector Machine (SVM), showcased the predictive power of implementing machine learning. Class membership for these behaviors (offender vs. non-offender) was predicted at rates well above baseline (e.g., 50% at baseline vs 87% accuracy for academic misconduct with just three input variables). This study demonstrated FoMO's relationships with these behaviors as well as how machine learning can provide additional predictive insights that would not be possible through inferential statistical modeling approaches typically employed in psychology, and more broadly, the social sciences. Research in the social sciences stands to gain from regularly utilizing the more traditional statistical approaches in tandem with machine learning.

**Data Availability Statement:** All relevant data and code will be made available at: https://osf.io/r7xyn/?view_only=8191203963dd46ae87996116102cf305.

**Funding:** This project was generously supported by the Dr. Marjy Ehmer Fund, of the psychology department at Southern Connecticut State University, awarded to PCM and CJB. The funders had no role in study design, data collection and analysis, decision to publish, or preparation of the manuscript.

**Competing interests:** The authors have declared that no competing interests exist.

# Introduction

Stuck finishing work with approaching deadlines you decline your colleagues' invitation to a local restaurant, but you feel uneasy that you are missing out on the fun. This uneasiness is the fear of missing out (FoMO). FoMO—chronic apprehension that one is missing rewarding/fun experiences peers are experiencing [1]—has gained considerable research and media attention. Although most prevalent between 18 and 34 years [2], only 13% of individuals report never experiencing FoMO [3]. FoMO positively associates with disruptive/harmful social media use and lower life satisfaction [1]. Better understanding how FoMO influences individual behavior and functioning will benefit the individual differences literature while contributing to interventions to reduce FoMO's negative influence. Therefore, we assessed relationships between FoMO and a broad array of maladaptive behaviors. Six hypotheses tested the relationship between FoMO, with relevant moderating demographic variables, and academic misconduct, drug use, alcohol use, and illegal behaviors.

Secondarily, we investigated benefits of machine learning approaches in tandem with traditional statistical modeling. This underutilized approach allows for inference via null hypothesis significance testing (i.e., traditional statistical analysis approaches; Part 1) and prediction via supervised machine learning (Part 2). Psychology often focuses on explaining causal relationships using traditional statistical approaches that can fall short of meaningfully predicting future behavior [4]. While statistics enables us to make inferences, causal, or associative claims about relationships, prediction enables forecasting outputs given an input, or set of inputs, with great accuracy that does not require prior assumptions. Prediction is the main goal of supervised machine learning [5]. To this end we asked two research questions examining if FoMO can predict behavior above chance, and if so, how much weight does it carry compared to other variables.

## College student behavior

College is a major transition that could facilitate psychological growth or maladaptive behaviors and psychological problems. Transitioning to college is a milestone; young adults leave their homes' safety and familiarity and step into the "unknown,"—an entirely novel environment both liberating and intimidating. To maintain wellbeing and motivation during this transition, self-determination theory (SDT) suggests that needs for autonomy, competence, and social relatedness require fulfillment. Autonomy is freedom to direct one's thoughts and behaviors toward valued goals, competence is the need to feel effective in domains important to self-identities, and social relatedness is a desire for close, warm, and trusting interpersonal relationships [6].

Yet increased autonomy could be challenging for students. Research indicates that college students who have "helicopter parents" (lack of autonomy) experience negative outcomes including lower social relatedness, competence, and autonomy fulfillment [7]. Alternatively, complete independence and autonomy might overwhelm some college students if they lack guidance and direction that parents provided previously. As such they may engage social comparison processes to determine appropriate behavior in this new context. Depending on the target of the social comparisons, college students could adopt common, but maladaptive behaviors (e.g., substance abuse, academic and criminal misconduct, risky sexual behaviors; [8–14].

## Relationships between FoMO and maladaptive behaviors in college

**Academic misconduct.**    While autonomy does relate to improved academic performance [15], students not performing well could be at risk for anxiety and depression [16,17]. For

higher FoMO students, anxiety could foster social comparisons that increase pressure to improve academic performance, perhaps via any necessary means. Per the Conservation of Resources (COR) [18] and Social Comparison Theories [19], college student depression and anxiety might be further exacerbated to the degree they socially compare their self to others they perceive as receiving greater resources, given that academic performance (i.e., grades) might be considered a career resource. Previous research indicates that aversive social comparisons and perceived resource loss can lead to unethical behaviors, including cheating [20]. Thus, underperforming students might be more likely to engage in cheating or other academic misconduct to increase their career resources and status when socially comparing themselves to others because underperformance could suggest a threat to competence need fulfillment as SDT suggests. Research already notes that college students with higher FoMO levels are more likely to use Facebook during classroom lectures (a form of academic incivility) [1]. It has also been found that males generally report higher levels of academic misconduct compared to females [21]. Thus:

H1: Higher FoMO levels will be associated with academic misconduct.

H2: Living situation (a), SES (b), and gender (c) will moderate the above relationship.

**Substance use.**   Although SDT argues that wellbeing results from autonomy, competence, and relatedness need fulfillment, college students may struggle to fulfill those needs. Besides academic misconduct, substance use is a romanticized part of the college experience [22] that leads to negative consequences. To reduce FoMO, students might use substances to "fit in" or belong in a peer group to fulfill social relatedness needs. Thus, student FoMO may predict substance use via social comparisons and baseline expectancies. For example, Riordan and colleagues [23] reported that high FoMO undergraduate students did not engage in more drinking overall, but they did consume a greater quantity in a single drinking episode and experienced more negative consequences. Additionally, Greek life/fraternity/sorority involvement increases college student alcohol use [24]. Given the ubiquity of cannabis and similarities in attitudes between that substance and alcohol, we also expected similar processes may contribute to increased drug use by college students. Illicit drug, nicotine, and alcohol use is much more prevalent in men than with women, although the relationship with alcohol seems to disappear among adolescents (ages 12–17) [25]. Therefore:

H3: FoMO will be associated with drinking behavior (a) and drug use (b).

H4: Living situation (a), SES (b), and gender (c) will moderate the above relationship.

**Illegal behaviors.**   Social comparisons and FoMO could also contribute to illegal behavior. Being with peers engaging in illegal activity may be perceived as less severe if one also has high FoMO. Per COR theory, the threat of being left out may be experienced as a threat to one's status, social relatedness, or reputational resources–needs requirement fulfillment for wellbeing and motivation per SDT. Although research is limited, some findings suggest that high FoMO individuals are more likely to engage in low-level illegal behavior such as driving while using a cell phone [1]. Therefore, to provide some evidence bearing on this potential relationship, we examined whether higher FoMO college students also reported engaging in a higher frequency of illegal behavior than their lower-FoMO counterparts. Moreover, gender is one of the strongest predictors of delinquency and violent criminal behavior with males being perpetrators at much higher rates than females [26,27]. As such:

H5: FoMO will be associated with illegal behavior.

H6: Living situation (a), SES (b), and gender (c) will moderate the above relationship.

## Prediction of maladaptive behaviors

Statistical modeling approaches (i.e., null hypothesis significance testing) draw inferences concerning relationships, whereas machine learning quantifies predictive values based on a

defined set of input variables (for an overview see Hastie and colleagues [28]). However, both approaches together can lead to richer insights than either alone.

Machine learning is a computer science subfield that builds algorithms which learn via data exposure without explicit instruction. The machine learning we employed, supervised learning, infers a function that maps inputs to outputs. This function (i.e., the model) allows predictions using the data. More specifically, the supervised learning algorithm divides the data into two sets: a "training" and a "test" set. The training set allows the algorithm to learn the relationship between input variables and the data's label to develop a model. The test set determines the algorithm's predictive power. The test set represents the data unseen in the training phase with a typical split being 80% of the data for the training set and the remaining 20% for the test set. To minimize the bias introduced by training and test set selection, k-fold cross-validation is often applied. The dataset is split into k folds, where the folds represent non-overlapping subsets, and k is typically in the range 5 to 20. The model is evaluated k times as follows: one of the folds is treated as the test set, and the remaining folds represent the training set. For k = 5, this scheme results in 5 evaluations corresponding to the 5 possible selections of test and training sets. The reported performance measure of the model is the average score across the 5 folds. This is termed cross-validation score.

Although most past work on FoMO (and in the broader social sciences) relied on traditional statistical modeling approaches, machine learning is starting to be adopted. Machine learning processes have elucidated the predictive validity of FoMO concerning problematic smartphone use [29]. Those authors entered several psychopathological and demographic variables to determine their ability to predict problematic smartphone use. They further discussed the compatibility of machine learning alongside theoretical frameworks in psychological research [30]. Additionally, neural networks and decision trees were used to predict sixth semester CGPA as a proxy for academic performance [31]. We utilized both modeling approaches (hierarchical linear regression: Part 1; machine learning: Part 2) to better understand FoMO's influence on maladaptive student behavior. This work expands our understanding of college student FoMO by leveraging complementary and convergent statistical and machine learning approaches. Therefore, in Part 1 we identify relationships via traditional methods (i.e., hierarchical linear regression) and in Part 2 we use machine learning to address two research questions that build off those previous hypotheses:

RQ1: If FoMO is found to have relationships with different maladaptive behaviors, can machine learning algorithms predict those behaviors in college students beyond random chance?

RQ2: If FoMO is found to have relationships with different maladaptive behaviors and machine learning algorithms can predict those behaviors in college students beyond random chance, how much predictive weight will FoMO carry compared to other demographic features?

Thus, we proceeded by evaluating FoMO as a predictor of college student maladaptive behavior in the form of drinking, drug use, and illegal behavior and stands among the small minority to utilize supervised machine learning in conjunction with statistical analysis in psychological research. Overall, the intent of the study is twofold: 1) to investigate the role of FoMO on maladaptive behaviors in college students and the moderating role of demographic variables through statistical modeling, and 2) quantify the predictive power of FoMO and these demographic variables through machine learning methods.

The differences between the two approaches employed in our paper have been a subject of some debate, so we include some brief comments to highlight these differences. For a more detailed treatment, the reader is directed to Bzdok and colleagues [32]. While machine learning is built on a statistical framework and often includes methods that are employed in

statistical modeling, its methods also draw on fields such as optimization, matrix algebra, and computational techniques in computer science. The primary difference between the two approaches is in how they are applied to a problem and what goals they achieve. Statistical inference is concerned with proving the relationship between data and the dependent variable to a degree of statistical significance, while the primary aim of machine learning is to obtain the best performing model to make repeatable predictions. This is achieved by using a test set of data as described earlier to infer how the algorithm would be expected to perform on future observations. When prediction is the goal, a large number of models are evaluated and the one with the best performance according to a metric of interest is deployed.

## Methods

### Participants and procedure

Four hundred and ninety undergraduate participants from a Northeastern university completed our cross-sectional survey. However, we excluded 18 participants that were not in the targeted age range (i.e.,18–24 years), leaving a final analyzed sample of $n$ = 472 participants with no missing item-level data ($M$age = 19.06, $SD$age = 1.17; 52% white, 23% black, 4% Asian, .2% Pacific Islander/Alaskan Native; 28% male). Measures within that survey assessed trait FoMO levels, drinking behaviors, drug use behaviors, and questions pertaining to unethical and illegal behavior while in college. All participants provided their written informed consent. This study was approved by the Southern Connecticut State University's Institutional Review Board (IRB).

## Measures

### Part 1: Traditional statistical modeling

*FoMO*. We used Pryzbylski et al.'s 10-item Fear of Missing Out scale [1]. Participants rated how the truth of each statement (e.g., "I fear my friends have more rewarding experiences than me" and "I get anxious when I don't know what my friends are up to.") in reference to the self on 1 (Not at all true of me) to 5 (Extremely true of me) scales. Higher mean scores represent higher levels of trait FoMO.

*Drinking and drug use*. This study used the Drinking and Drug Habits Questionnaire. (DDHQ) [33]. The DDHQ is a 13-item, self-report frequency measure of usage across many drug classes. Participants reported whether they had ever used each substance, specifically since entering college. The drug classes were: marijuana, "powder" cocaine, "crack" cocaine, amphetamines (speed), methamphetamine (Meth), opiates (heroin, etc.), pain medications used for non-medical purposes (Oxycontin, Percocet, etc.), Methadone, barbiturates (downers), tranquilizers (Valium, Xanax, etc.), hallucinogens (LSD, mushrooms, etc.), "club drugs" (ecstasy, ketamine, etc.), inhalants (paint, fumes, etc.), and other non-pain killer prescription medications sued for non-medical purposes (Ritalin, Adderall, etc.).

*Unethical and illegal behavior questionnaire*. This study employed a self-report questionnaire assessing unethical and/or illegal behaviors relevant to the college setting. Participants anonymously reported whether they had ever engaged in nine different behaviors, since entering college. Those included: stealing, physical fighting, selling illegal drugs, giving away illegal drugs, selling their own prescription medications, giving away prescription medications, academic cheating, plagiarism, and receiving formal college disciplinary action.

### Data analysis

**Part 1: Traditional statistical modeling.**  All statistical analyses were run by IBM SPSS Version 26.0 statistical software package. A series of hierarchical regression analyses were

conducted to test the association between trait level FoMO and engagement in a broad range of maladaptive behaviors during college. For each dependent variable of interest, there were three separate regression models run. On Step 1, an alternating demographic variable of interest (gender, socioeconomic status, living situation) and FoMO were entered. We dummy-coded living situations (living with parents = 0) for analysis. To test for a potential interaction of trait FoMO and demographic on the criterion variables, FoMO X Demographic was entered at Step 2 of the regression models. Note, not all possible outcome variables included in the measures (e.g., all illegal behaviors, all drug classes) were analyzed as part of the hypothesis testing. Nonetheless, we included them in the correlation tables so that future research may use any potential information as a foundation for hypothesis or exploratory testing. Given the number of tests we report, we also have truncated several of the results reports to the most pertinent statistical information. Full model results for all statistical tests can be viewed in the online supplemental material.

## Part 2: Machine learning

*FoMO*. We used two different approaches to examine FoMO's predictive value with regards to maladaptive behavior class membership. The first approach utilized the mean FoMO aggregated across all 10 items as the predictor variable. The second approach used each individual question's score instead of the mean as the predictor variables.

*Framing maladaptive behavior as a binary classification problem*. We determined that collapsing each maladaptive behavior into a binary (non-offender/offender, nondrinker/drinker, etc.) classification problem was the most meaningful for predicting behaviors, as well as showcasing the utility of machine learning. While clinical diagnosis is slowly moving toward more dimensional approaches, diagnostic classification remains the long-established norm, especially in clinical practice [34]. Hence, as an initial analysis it was preferable to use the binary classifications that are typically clinically used. Future research can investigate more nuanced and specific expanded classification problems (e.g., nonuser/experimenter/heavy drug user). It's important to note that in the case of a balanced dataset a binary classifier at baseline can achieve a 50% prediction accuracy by always predicting the same class.

*Alcohol*. Based on DDQ past month drinking frequency. Class 0: Non-drinker/light drinker, "I do not drink at all" or "About one per month". Class 1—Moderate/Heavy drinker,"2–3" times a month", "3–4" times a month or "Nearly every day" or "once a day or more", all remaining participants.

*Drugs*. Based on several drugs without cannabis due to its ubiquitous and legal nature in many places. Drugs included are cocaine (power and crack), amphetamines, methamphetamines, opiates, pain medications, methadone, barbiturates, tranquilizers, hallucinogens, club drugs, inhalants, and prescription stimulants. All scores for these questions were summed. Class 0—Nonuser, no use on any of the drugs. Class 1—User, all remaining participants.

*Academic misconduct*. Based on plagiarism and cheating responses. All scores for these questions were summed. Class 0—Nonoffender, total score equals 0. Class 1—Offender, all remaining participants.

*Illegal behavior*. Based on several illegal behavior questions. Illegal behaviors included stealing, physical fighting, speeding, reckless driving, driving under the influence (DUI), selling illegal drugs, giving away illegal drugs, selling prescription medication, and giving away prescription medication. All scores for these questions were summed. Class 0—Nonoffender, total score equals 0. Class 1—Offender, all remaining participants.

All analyses were run using the Python machine learning library scikit-learn through Jupyter Notebooks. For experiment reproducibility, we used a fixed random seed for the selection

of the training and test sets. A training sample of 75% (N = 354) was randomly selected with the remaining 25% (N = 118) set aside for the test sample.

The machine learning classifiers included Support Vector Machine (SVM) using two kernel functions, linear and Radial Basis Function (RBF), decision trees, random forests, and logistic regression. A discussion of the relative merits of various classifiers and the modeling tradeoffs involved in each is beyond the scope of this article. Interested readers are directed to review Kotsiantis [35] for a detailed survey of common machine learning algorithms. In the remainder of this section, we briefly summarize algorithms that we used in our analysis.

Decision trees are supervised machine learning classifiers that filter data in the likeness of trees: Roots to branches to leaf nodes. Using if-then sorting, it classifies by categorizing data into progressively smaller sub-categories like trunk to branches and then leaves.

Random forest classifiers are an ensemble of individual decision trees working together, to provide the best predictive model based on majority group consensus.

Support Vector Machines (SVM) are classifiers that assign the best hyperplane that distinguishes between possible classes. SVM algorithms are especially useful and achieve greater predictive accuracy when the classes are not linearly separable. It is important to note that while SVM is technically a linear classifier, the use of the Radial Basis Function (RBF) kernel allows data to be classified when the relationship is nonlinear. The RBF kernel is also referred to as the "kernel trick".

Logistic regression classifiers are a familiar concept stemming from traditional statistics in which the probability of the default class is modeled using a sigmoid function. Probability values are then converted into either of the two class labels using a thresholding approach.

In addition to just offering predictive value from the input variables we provide; machine learning models grant the ability of feature selection. Feature selection is the ability of the models to automatically select the best set of features from the data set that maximize predictive power while reducing the number of variables included. We also use principal component analysis (PCA) and recursive feature elimination (RFE) for dimensionality reduction. To explore the merit of dimensionality reduction techniques we also applied RFE and PCA in combination with a random forest classifier.

Whenever applicable, we used the grid search technique to optimize the model hyperparameters. Model results were compared by prediction accuracy, F1-score, and ROC AUC score. While there are a few metrics that can be considered when evaluating machine learning results, we focus on just three: accuracy, F1-score, and ROC AUC score. Accuracy is the metric that reports the percentage of all cases identified correctly. If we had five cases and four were correctly identified, the model would have an accuracy of 80%. Note that in this paper accuracy scores are reported as a fractional value (i.e., .80). Accuracy is most appropriate for when all cases are equally weighted in importance or when the class distributions are similar. When the importance of all cases is not of equal weight or the cases are not similarly distributed, the F1-score is more meaningful. The F1-score provides a better metric that incorporates cases incorrectly classified. The F1-score does this by being the harmonic mean of precision, percentage of correctly identified positive cases from all cases predicted as positive, and recall, the percentage of correctly identified positive cases from all cases that are actually positive. The ROC AUC score is another widely used metric for evaluating the skill of a prediction model. ROC, which is short for receiver operating characteristic curve, is a function that captures the relationship between the false positive rate and the true positive rate of a classifier for varying threshold values, where the threshold is used to map probabilities to a class label. As such, the ROC curve makes it possible to calibrate the threshold to achieve the best balance between the true positive rate and the false positive rate. The ROC AUC score is the area under the ROC curve. An ROC AUC score of 0.5 corresponds to a no-skill classifier whereas a score of 1.0 is

that of a perfect-skill classifier. The machine learning approaches were used to predict each of the dependent variables based on the aggregate FoMO measure and individual FoMO scale items.

In addition to the evaluative measures such as accuracy, F1-score and ROC AUC score, machine learning models make it possible to derive feature importance scores. Feature importance represents techniques that produce scores of input features that denote their utility for predicting the dependent variable. Feature importance scores can provide insights into the dataset and the model that can guide the researcher in the optimization of the model and the collection of further data.

For each dependent variable (i.e., alcohol, drug, academic, or illegal) the methodology was the same and as follows. After identifying and labeling our dependent variable, we applied SVM using linear and RBF kernel functions. Minority classes were upscaled to have the same sample size as the majority class. A grid search was run to find the RBF-based SVM model with the best hyperparameter combination. The best parameters found were then implemented by the model to generate predictions. A grid search was run to find the linear SVM model with the best hyperparameter combination. The best hyperparameter values found were then employed by the model to generate predictions. Feature importances for a decision tree model (criterion = entropy, max tree depth = 4) were derived to determine signal size of each predictor variable. The same was done with a random forest classifier (number of estimators = 100, max depth = 4). In addition to the base random forest model, RFE was used in combination with random forest to reduce the dimensionality of the dataset. RFE selected the specified number of best features that gave the best performance for the estimator (random forest). Principal component analysis (PCA) was used to create k features that were linear combinations of predictor variables. For aggregate FoMO, RFE selected 2 out of the 4 original features and PCA reduced the dataset to two principal components. For individual FoMO items, RFE selected 4 out of the 13 original features and PCA reduced the dataset to four principal components. Logistic regression was used as the final comparative model for the binary conditions (collapsed classes).

## Results

### Part 1—traditional statistical modeling

Analysis of the correlation between FoMO and maladaptive behaviors reveals interesting relationships across all four domains. Regarding academic misconduct, higher levels of FoMO were found to be correlated with higher rates of classroom incivility and plagiarism. Greater typical weekly alcohol consumption and a lower age when first beginning drinking alcohol were also correlated with increased levels of FoMO. Additionally, FoMO is correlated with increased cannabis, stimulant, depressants, and hallucinogen use. Finally, when looking at illegal behaviors, FoMO had positive correlations with stealing, giving away illegal drugs, and giving away prescription medication. See Table 1 for full results.

### Hypothesis testing

*Academic misconduct.* Contrary to expectations, the interaction between FoMO and gender did not contribute unique variance when predicting classroom incivility. Therefore, we dropped the interaction from analysis. The results showed that FoMO was positively associated with classroom incivility as did being male. When examining living situations and FoMO as predictors of classroom incivility neither living situation nor the FoMO by living situations interaction were significant predictors. Still, higher FoMO levels significantly predicted more classroom incivility. Next, we examined whether FoMO and SES interacted to predict

**Table 1. Correlations and descriptive statistics among all measures.**

| Instruments/Subscales | 1 | 2 | 3 | 4 | 5 | 6 | 7 | 8 | 9 | 10 | 11 | 12 | 13 | 14 | 15 | 16 | 17 | 18 |
|---|---|---|---|---|---|---|---|---|---|---|---|---|---|---|---|---|---|---|
| 1. FoMO | - | | | | | | | | | | | | | | | | | |
| 2. Classroom Incivility | *.281* | - | | | | | | | | | | | | | | | | |
| 3. Cheating | .077 | *.341* | - | | | | | | | | | | | | | | | |
| 4. Plagiarism | *.136* | *.249* | *.398* | - | | | | | | | | | | | | | | |
| 5. Typical Weekly Alcohol | *.207* | *.171* | *.186* | *.189* | - | | | | | | | | | | | | | |
| 6. Age Began Drinking | *-.142* | *-.118* | .018 | .033 | *-.228* | - | | | | | | | | | | | | |
| 7. Cannabis | *.124* | *.197* | *.144* | .082 | *.374* | *-.281* | - | | | | | | | | | | | |
| 8. Stimulants | *.187* | *.165* | *.133* | *.193* | *.511* | *-.206* | *.276* | - | | | | | | | | | | |
| 9. Depressants | *.161* | .075 | .040 | .067 | *.222* | -.076 | *.148* | *.462* | - | | | | | | | | | |
| 10. Hallucinogens | *.091* | .076 | .034 | .056 | *.477* | *-.237* | *.319* | *.557* | *.337* | - | | | | | | | | |
| 11. Stealing | *.172* | *.198* | *.266* | *.134* | *.262* | -.102 | *.327* | *.248* | *.156* | *.214* | - | | | | | | | |
| 12. Physical Fighting | .025 | *.122* | *.105* | *.118* | *.251* | -.047 | *.157* | *.170* | *.109* | *.242* | *.215* | - | | | | | | |
| 13. Speeding | -.045 | -.014 | *.156* | .027 | .051 | .044 | .070 | .007 | .077 | -.025 | -.051 | .060 | - | | | | | |
| 14. Reckless Driving | -.036 | .006 | *.108* | *.116* | .000 | .065 | -0.16 | -.013 | *.115* | -.031 | -.055 | .023 | *.479* | - | | | | |
| 15. Selling Illegal Drugs | .075 | *.102* | *.151* | *.196* | *.311* | -.068 | *.287* | *.290* | .028 | *.317* | *.232* | *.260* | .055 | .022 | - | | | |
| 16. Giving Illegal Drugs | *.144* | *.190* | *.194* | *.207* | *.301* | -.099 | *.368* | *.295* | *.142* | *.294* | *.270* | *.177* | .028 | -.059 | *.437* | - | | |
| 17. Selling Rx Medication | .036 | .111 | .042 | .063 | *.288* | -.118 | *.121* | *.371* | *.124* | *.363* | *.118* | *.173* | .000 | -.022 | *.213* | .117 | - | |
| 18. Giving Rx Medication | *.095* | .076 | .031 | *.101* | *.273* | *-.124* | *.131* | *.449* | *.242* | *.215* | *.182* | *.260* | -.064 | -.033 | *.130* | *.226* | *.513* | - |
| Mean | 2.17 | 1.70 | 0.94 | 0.31 | 4.02 | 16.32 | 2.36 | 5.24 | 6.18 | 1.10 | 0.51 | 0.16 | 0.19 | 0.05 | 0.15 | 0.32 | 0.03 | 0.07 |
| Standard Deviation | 0.83 | 0.51 | 1.21 | 0.82 | 7.10 | 1.70 | 1.11 | 0.76 | 0.60 | 0.35 | 0.98 | 0.53 | 0.52 | 0.25 | 0.64 | 0.87 | 0.29 | 0.39 |

*Note.* Chronbach's Alpha for FoMO and Classroom Incivility are 0.894 and 0.856, respectively. Coefficients significant at $p < .05$ in bold. Coefficients significant at $p < .01$ in bold italics.

classroom incivility. The FoMO by SES interaction again did not explain unique variance, nor did SES uniquely predict incivility. Only FoMO significantly positively predicted classroom incivility.

Next, we examined interactions between FoMO, gender, living situations, and SES on plagiarism while in college. The FoMO by gender interaction failed to contribute unique variance to the model. However, both higher FoMO and being male predicted higher plagiarism self-reports. Like gender, living situations did not contribute unique model variance. In this model only FoMO and living off-campus (compared to living with parents) resulted in higher self-reports of plagiarism in colleges; no other living situations were significantly different from living with parents. Lastly, we examined FoMO and SES's contributions to plagiarism. The interaction qualified a significant FoMO effect; SES was non-significant. The contribution of FoMO to plagiarism was stronger at low SES than at average SES, but that relationship attenuated at high SES.

The final academic misconduct outcome we examined was self-reported cheating. FoMO and gender did not significantly interact on cheating; males reported more cheating whereas FoMO was not a significant cheating predictor. Similarly, living situation and FoMO did not significantly interact with cheating, and only living off-campus versus with parents resulted in more cheating. The full model examining FoMO and SES's influence on cheating was not significant. Together these results provide support for H1 and H2b, although we could not reject the null hypothesis for H2a and H2c. See Table 2 for a summary of found relationships. See supplemental materials for full results.

*Alcohol*. The FoMO by gender interaction did not predict weekly alcohol consumption. Yet being higher in FoMO and male both predicted higher weekly alcohol consumption. When

**Table 2. Summary of significant relationships from hierarchical regression modeling.**

| Domain | Maladaptive Behavior | Significant Predictor | b | p |
|---|---|---|---|---|
| Academic | Classroom Incivility | FoMO | 0.173 | < .001 |
| | | Gender | -0.102 | 0.044 |
| | Plagiarism | FoMO | 0.133 | 0.003 |
| | | Gender | -0.193 | 0.021 |
| | | FoMO X Low SES | 0.215 | < .001 |
| | | FoMO X Avg SES | 0.132 | 0.003 |
| | Cheating | Gender | -0.278 | 0.025 |
| | | Living off-campus vs. parents | 0.606 | 0.003 |
| Alcohol | Weekly Consumption | FoMO | 1.760 | < .001 |
| | | Gender | -1.745 | 0.015 |
| | | Living residence hall vs. parents | 1.700 | 0.013 |
| | | Living off-campus vs. parents | 3.382 | 0.004 |
| Drugs | Depressant Use | FoMO | 0.034 | < .001 |
| | | Gender | -0.037 | 0.016 |
| | | Living off-campus vs. parents | 0.073 | 0.003 |
| | Stimulant Use | FoMO | 0.019 | < .001 |
| | | Living off-campus vs. parents | 0.050 | 0.003 |
| | | Living other vs. parents | 0.148 | 0.009 |
| | Cannabis Use | FoMO | 0.165 | 0.007 |
| | | Living residence hall vs. parents | 0.474 | < .001 |
| | | Living off-campus vs. parents | 0.628 | 0.001 |
| | | Living other vs. parents | 1.299 | 0.038 |
| Illegal | Stealing | FoMO | 0.202 | < .001 |
| | | Living residence hall vs. parents | 0.271 | 0.004 |
| | | Living off-campus vs. parents | 0.511 | 0.002 |
| | | FoMO X High SES | 0.309 | < .001 |
| | | FoMO X Avg SES | 0.206 | < .001 |
| | Giving Away Illegal Drugs | FoMO | 0.150 | 0.002 |
| | | Gender | -0.322 | < .001 |
| | | Living off-campus vs. parents | 0.289 | 0.047 |
| | Giving Away Rx Drugs | FoMO | 0.044 | 0.002 |
| | | FoMO X Living residence hall | -0.137 | 0.003 |

Note. Only the found statistically significant relationships are reported in this table for each respective domain of maladaptive behavior. Please see supplemental materials for full results.

testing for the FoMO by living situation on weekly consumption, the interaction failed to contribute unique variance to the model and thus we interpreted the model without the interaction term. Those results indicated that higher FoMO and living in a residence hall, or off-campus compared to living with parents resulted in significantly higher average weekly alcohol consumption; no significant differences emerged between living with parents and other living situations. Lastly, FoMO and SES did not interact on weekly alcohol consumption, nor did SES uniquely contribute to predicting consumption; only FoMO significantly positively associated with consumption. Thus, we were unable to reject the null hypothesis for H2a, H2b, or H3c, but found support for H1.

*Drugs*. When examining whether FoMO and gender jointly influence depressant use we observed a non-significant interaction. However, the model excluding that interaction term

found that both higher FoMO and male gender both uniquely predicted higher depressant use in college students. Similarly, FoMO and living situations did not show a significant interactive influence on depressant use. When excluding that non-significant interaction from the model, the results showed that higher FoMO and living off-campus relative to with parents each predicted higher depressant use; all other living situation comparisons were no-significant. The last predictors of depressant use we tested were FoMO and SES, which did not significantly interact. When predicting that outcome without the interaction, only FoMO exhibited a significant positive relationship with college student depressant use; SES did not predict depressant use in students.

Next, we turned our attention to stimulant use as the focal outcome. FoMO and gender did not significantly interact on college student stimulant use. The model without that term indicated that only increased FoMO resulted in increased stimulant use by students. Gender did not predict stimulant use with this sample. When examining the interaction of FoMO with living situations we also did not observe a significant increase in variance explained, although higher FoMO, living off-campus and other living situations (versus living with parents) each predicted significantly higher stimulant use. We observed no differences in stimulant use for those living in residence halls and with parents. Lastly, we examined FoMO's interaction with SES on stimulant use by college students. This analysis too indicated a non-significant contribution by the interaction term. Like above, higher FoMO resulted in higher stimulant use although SES did not.

Lastly, we tested whether FoMO and the demographic variables interacted to predict cannabis use. FoMO and gender did not significantly interact on cannabis use; however higher FoMO did predict higher cannabis use whereas gender did not predict use. FoMO also did not significantly interact with living situations to predict cannabis use. Yet, FoMO and living situation did additively predict cannabis use with higher FoMO levels, and (compared to living with parents) living in residence halls, off campus, and in other arrangements each predicting increased cannabis use. FoMO and SES also failed to significantly interact on cannabis use, although FoMO significantly positively predicted cannabis use, SES did not. Taken together we found support for H1, but could not reject the null hypothesis for H2a, H2b, or H2c.

*Illegal behaviors*. FoMO and gender did not interact on stealing in college, nor did gender contribute unique variance although FoMO significantly positively associated with stealing. When examining the influence of FoMO and living situation on student theft during college, the interaction failed to contribute unique model variance. The simpler model without the interaction term showed that higher FoMO, and living in residence halls or off-campus (compared to living with parents) each contributed to increased stealing while in college; significant differences were not observed for living in other arrangements versus living with parents. FoMO and SES also interacted to significantly predict stealing in college. The strongest relationship between FoMO and stealing was at higher SES followed by average SES with that relationship attenuating at lower SES.

Next, we examined self-reported giving away of illegal drugs while in college. The FoMO by gender interaction did not contribute unique variance to this model; higher FoMO and male individuals both reported more instances of giving away illegal drugs. Similarly living situations and FoMO did not interact on giving away illegal drugs, but again higher FoMO and living off-campus (versus with parents) both resulted in more frequent giving away of illegal drugs; all other comparisons were non-significant. When examining SES, the interaction again was non-significant as was the unique contribution of SES; higher FoMO predicted higher rates of giving out drugs.

The last illegal behavior we examined was giving out prescription drugs while in college. FoMO and gender did not interact on this behavior, nor did gender directly predict giving

**Table 3. Performance metrics across models and behaviors for aggregate FoMO.**

| | Aggregate | | | | | | | | | | | |
|---|---|---|---|---|---|---|---|---|---|---|---|---|
| | Academic Misconduct | | | Alcohol | | | Illegal Behavior | | | Drugs | | |
| | Acc. | F1 | ROC | Acc. | F1 | ROC | Acc. | F1 | ROC | Acc. | F1 | ROC |
| SVM—RBF | 0.50 | 0.56 | 0.59 | 0.69 | 0.69 | 0.70 | 0.55 | 0.58 | 0.68 | 0.66 | 0.69 | 0.71 |
| SVM Linear | 0.47 | 0.53 | 0.60 | 0.72 | 0.72 | 0.70 | 0.64 | 0.67 | 0.68 | 0.66 | 0.69 | 0.71 |
| Decision Tree | 0.52 | 0.58 | 0.61 | 0.64 | 0.64 | 0.67 | 0.60 | 0.62 | 0.69 | 0.68 | 0.70 | 0.72 |
| RF | 0.54 | 0.60 | 0.69 | 0.73 | 0.73 | 0.75 | 0.60 | 0.63 | 0.75 | 0.66 | 0.69 | 0.77 |
| RFE + RF | 0.87 | 0.81 | 0.50 | 0.64 | 0.62 | 0.66 | 0.75 | 0.65 | 0.57 | 0.77 | 0.70 | 0.61 |
| PCA + RF | 0.87 | 0.81 | 0.44 | 0.61 | 0.58 | 0.65 | 0.75 | 0.65 | 0.55 | 0.78 | 0.70 | 0.62 |
| Logist. Reg. | 0.54 | 0.67 | 0.48 | 0.73 | 0.69 | 0.72 | 0.63 | 0.71 | 0.64 | 0.64 | 0.39 | 0.61 |

Note. Performance metrics (accuracy, F1-score and ROC AUC) obtained from each of the machine learning models across behavior domains using the mean score across all FoMO items as a predictor. The methods that combined a dimensionality reduction approach with random forests (RFE + RF and PCA + RF), achieved the highest accuracy for all behavior domains with the exception of alcohol consumption.

away prescription drugs. Still, higher FoMO did predict higher rates of prescription drug giving. However, FoMO did interact with living situations to predict giving away prescription drugs. Specifically, living in residence halls compared to living with parents resulted in less giving away of prescription drugs. Neither FoMO's interaction with living off-campus nor living in other situations significantly predicted giving away prescription drugs in college. Lastly, the model examining the joint influence of FoMO and SES failed to achieve statistical significance. The results provide support for H1, H2a, and H2b, while we could not reject the null hypothesis for H2c.

## Part 2- machine learning

In Tables 3 and 4 below, we show the results of applying the classifiers to predict the four variables of interest. For each of the measures, we show the achieved accuracy, F1-score, and ROC AUC (denoted by ROC in the table header) using the two modeling scenarios described earlier, denoted as "Aggregate" and "Individual" in the tables. The former refers to using just the mean score across all ten FoMO items as a single predictor while the latter refers to using the

**Table 4. Performance metrics across models and behaviors for individual FoMO items.**

| | Individual | | | | | | | | | | | |
|---|---|---|---|---|---|---|---|---|---|---|---|---|
| | Academic Misconduct | | | Alcohol | | | Illegal Behavior | | | Drugs | | |
| | Acc. | F1 | ROC | Acc. | F1 | ROC | Acc. | F1 | ROC | Acc. | F1 | ROC |
| SVM—RBF | 0.53 | 0.59 | 0.58 | 0.69 | 0.68 | 0.53 | 0.59 | 0.62 | 0.65 | 0.70 | 0.71 | 0.64 |
| SVM Linear | 0.48 | 0.54 | 0.60 | 0.74 | 0.74 | 0.69 | 0.53 | 0.56 | 0.65 | 0.63 | 0.66 | 0.69 |
| Decision Tree | 0.58 | 0.63 | 0.63 | 0.64 | 0.63 | 0.66 | 0.57 | 0.60 | 0.68 | 0.71 | 0.73 | 0.81 |
| RF | 0.60 | 0.65 | 0.80 | 0.69 | 0.69 | 0.79 | 0.59 | 0.61 | 0.80 | 0.71 | 0.73 | 0.57 |
| RFE + RF | 0.87 | 0.82 | 0.51 | 0.61 | 0.57 | 0.61 | 0.74 | 0.65 | 0.55 | 0.77 | 0.71 | 0.60 |
| PCA + RF | 0.87 | 0.81 | 0.54 | 0.62 | 0.59 | 0.62 | 0.75 | 0.65 | 0.55 | 0.78 | 0.70 | 0.62 |
| Logist. Reg. | 0.48 | 0.62 | 0.43 | 0.73 | 0.70 | 0.73 | 0.59 | 0.69 | 0.57 | 0.65 | 0.42 | 0.63 |

Note. Performance metrics (accuracy, F1-score and ROC AUC) obtained from each of the machine learning models across behavior domains using the individual FoMO items as predictors. Consistent with the aggregate FoMO scenario, those models that combined a dimensionality reduction techniques with random forests (RFE + RF and PCA + RF) achieved the highest accuracy for all behavior domains with the exception of alcohol consumption. Using the individual scores does not appear to improve the model predictions compared to the aggregate scenario.

score of each of the ten items as separate predictors. Table 3 shows the performance metrics obtained for each of the classifiers employed in our analysis for the aggregate modeling scenario, while Table 4 shows the results for the individual scenario. The first two classifiers, labeled "SVM—RBF" and "SVM Linear" are variations of support vector machines. The first variation can model non-linearity in data, while "SVM Linear" is more appropriate for linearly-separable data. The next two classifiers, labeled "Decision Tree" and "RF" (short for random forest), are tree based models. Decision trees, while not generally the best performing models, are highly interpretable. Random forests are generally superior to decision trees and are more robust to overfitting. The next two techniques, "RFE + RF" and "PCA + RF", combine two different dimensionality reduction techniques, namely recursive feature elimination (RFE) and principal component analysis (PCA), with random forests. The last model shown is logistic regression ("Logist. Reg." in the tables).

The classifiers utilized, while by no means comprehensive, embody a range of disparate approaches that make it possible to draw reasonable conclusions as to how well FoMO items can predict each of the behavior domains. The three metrics–accuracy, F1-score and ROC AUC (denoted as ROC in the table header)–reveal different aspects of the skillfulness of the classifier. The range of the values for the metrics is [0, 1], where 0 indicates a no-skill model, and 1 represents perfect skill. The question of which metric is most informative is to a great extent dependent on the goal of the modeling scenario and the characteristics of the dataset. For instance, as discussed earlier, accuracy, being the percentage of instances that the classifier predicts correctly, is not very informative if false negatives or false positives carry different weights, or the dataset is imbalanced. In such cases, the F1-score is a much more appropriate metric. On the other hand, while ROC AUC is a widely adopted general metric for evaluating algorithm performance, it may not be an informative metric for evaluating a model that is calibrated using a given decision threshold. Hence, we include all three metrics in Tables 3 and 4 to allow for a broad assessment of the performance of the techniques employed.

In interpreting the results in Tables 3 and 4, it is useful to not only focus on the value of a given metric but observe the level of concordance among the 3 metrics for a given classifier and domain of behavior. The higher the metric values are and the level of agreement between them, the more performant the model is in the broad sense. Examples of such a pattern can be observed for several classifiers used to predict drug use, alcohol and illegal behavior, in both the individual and aggregate scenarios (e.g. Aggregate / Decision Tree / Drugs; Aggregate / RF / Alcohol; Individual / SVM Linear / Alcohol). When the metrics do not have a high level of agreement, however, a more nuanced interpretation is warranted. For instance, for the case of the RFE + RF classifier for predicting academic misconduct in the aggregate scenario, the difference between the accuracy (0.87) and F1-score (0.81) on the one hand, and ROC AUC (0.50) is substantial. Such a result indicates that while the selected model is reasonably skillful, other choices of decision thresholds for model calibration might result in a degraded performance. For the purposes of this paper, since our goal is to demonstrate the effectiveness of machine learning approaches for predicting domains of behavior in general, we focus on accuracy as the metric of choice. Fig 1 shows the accuracy scores from Tables 3 and 4, grouped by classifier and modeling scenario (individual vs. aggregate). It is worth noting that a comparison of model performance based on the other two metrics would only be meaningful if the application scenario is known and the relative costs of false positives and false negatives can be assessed.

Among the approaches considered, those that combined dimensionality reduction with a machine learning method, namely PCA and RFE combined with random forest, produced the highest accuracy for 3 out of 4 measures of interest in the case of using aggregate FoMO measure (FoMO mean) as an input variable. These are Academic Misconduct, Illegal Behavior,

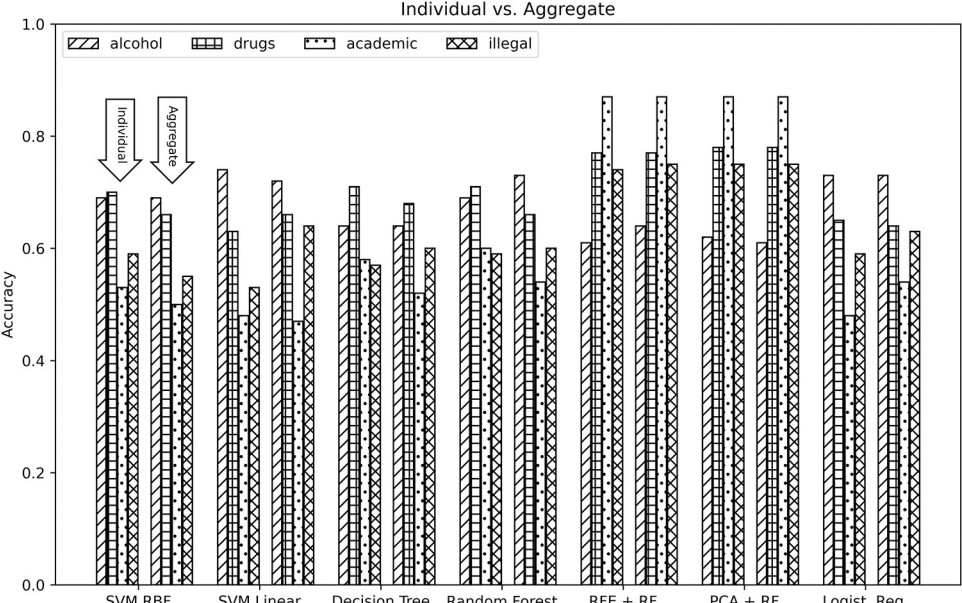

**Fig 1. Comparing aggregate and individual FoMO item performance metrics.** Comparison of model accuracy for the aggregate scenario vs. the individual scenario across behavior domains based on values shown in Tables 3 and 4. Results are aggregated by machine learning model and scenario, with solid bars representing accuracy values for the aggregate scenario and the bars with patterns showing the results for the individual scenario.

and Drugs with accuracy scores of 0.87, 0.75, and 0.78, respectively (See Table 3). For Alcohol, random forest produced the best result among all approaches considered, with an accuracy of 0.73. For both RFE and PCA we used 2 as the number of dimensions, down from 4. When considering the individual FoMO measures as input variables, the highest accuracy values resulted from the same models as in the aggregate case. Further, these accuracy values are comparable to the aggregate case for all outcome measures. For the models that included RFE and PCA, we reduced the number of variables to 4 from the original 13. When comparing the results from the models that included individual FoMO variables to those that used the aggregate FoMO measure instead, we can conclude that the former does not carry an advantage in terms of predictive power. This lends support to the notion that the aggregate FoMO measure is a robust indicator of trait FoMO levels.

As an illustrative example of how a prediction is carried out, Fig 2 shows the decision tree for drug offense/use classification based on the aggregate scenario. Although decision trees are typically not the best performing models, they allow for clear interpretability, a characteristic that other models trade off for higher predictive power. Starting with the root of the tree in Fig 2, the first decision that the tree uses to predict class membership is FoMO score. If the FoMO score is greater than 2.55, the subject is always predicted as an offender/user, independent of all other factors. When the FoMO score does not exceed the 2.55 threshold, it is still possible to be predicted as an offender/user, however, it is not as likely as being a nonoffender/nonuser. Being on the lower end of the FoMO scale is where living situation mattered in predicting class membership. The same pattern can be observed in the trees corresponding to other maladaptive behaviors (shown in the Supplemental Materials). For all cases, the decision at the root node is based on a FoMO score threshold, which results in a strong separation in class membership. Demographic profiles were only meaningful predictors for lower FoMO scores.

In addition to metrics such as accuracy, F1-score and ROC AUC, we compute feature importance scores from the models considered. An importance score is a measure of the

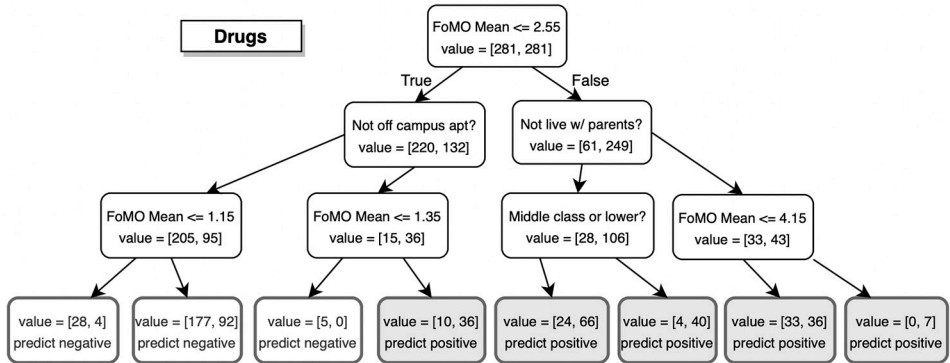

**Fig 2. Decision tree output for drug use.** Decision tree for drug offense/use classification based on the FoMO aggregate scenario. Starting at the root node, an example is evaluated in a sequential manner down the tree based on the conditions in the decision nodes. A classification is made according to the end node reached (blue denotes a positive prediction and light orange a negative prediction).

individual contribution of the feature to the classifier. The higher the score, the higher the contribution to the model. Importance scores can be used to guide feature selection for a more compact model, which in some cases improves model performance and algorithm efficiency. In Table 5, we show mean feature importance scores across all models. The results indicate that FoMO aggregate has by far the highest predictive value across all target variables. The mean feature importance scores for FoMO aggregate across all models considered are as follows (from highest to lowest): Drugs (0.63), Alcohol (0.63), Academic Misconduct (0.55), Illegal Behavior (0.49). Of the 3 non-FoMO variables, Living Situation carries the highest predictive value for all target variables except for Academic Misconduct (score = 0.03).

**Table 5. Mean feature importance scores across behaviors.**

| | Mean Feature Importance | | | |
|---|---|---|---|---|
| | **Academic Misconduct** | **Alcohol** | **Illegal Behavior** | **Drugs** |
| Gender | 0.17 | 0.05 | 0.07 | 0.04 |
| Living Situation | 0.05 | 0.19 | 0.24 | 0.15 |
| SES | 0.10 | 0.06 | 0.08 | 0.09 |
| FoMO Mean | 0.55 | 0.63 | 0.49 | 0.63 |
| FoMO1 | 0.04 | 0.03 | 0.11 | 0.07 |
| FoMO2 | 0.04 | 0.08 | 0.02 | 0.09 |
| FoMO3 | 0.02 | 0.03 | 0.06 | 0.12 |
| FoMO4 | 0.01 | 0.25 | 0.02 | 0.07 |
| FoMO5 | 0.18 | 0.05 | 0.04 | 0.28 |
| FoMO6 | 0.06 | 0.06 | 0.26 | 0.09 |
| FoMO7 | 0.04 | 0.03 | 0.11 | 0.03 |
| FoMO8 | 0.24 | 0.14 | 0.03 | 0.05 |
| FoMO9 | 0.17 | 0.12 | 0.11 | 0.04 |
| FoMO10 | 0.05 | 0.09 | 0.05 | 0.04 |

Note. Mean feature importance scores obtained from machine learning models considered across behavior domains. An importance score measures the individual contribution of the feature to the classifier. The higher the score, the higher the contribution to the model. The aggregate FoMO metric (denoted 'FoMO Mean' in the table) has a substantially higher importance score than all other predictors across all behavior domains. When considering the individual scenario, importance scores for FoMO items vary substantially across behavior domains. For instance, 'FoMO 8' has an importance score of 0.24 with respect to academic misconduct but only 0.03 for illegal behavior.

Gender, on the other hand, has a very low predictive value among non-FoMO variables except for Academic Misconduct. When considering the individual FoMO items, different items carry the strongest signal in relation to each of the four target variables. The FoMO items with the highest average importance scores relative to the dependent variables are as follows: FoMO6 ("Sometimes I wonder if I spend too much time keeping up with what's going on") for Illegal Behavior (0.26), FoMO5 ("It is important that I understand my friends "in jokes"") for Drugs (0.28), FoMO4 ("I get anxious when I don't know what my friends are up to") for Alcohol (0.25), FoMO8 ("When I have a good time it is important for me to share the details online —e.g., updating status") for Academic Misconduct (0.24).

## Discussion

### Summary

This study examined the relationship of trait level FoMO in college students and engagement in maladaptive behaviors through the lens of traditional statistical modeling and supervised machine learning. Overall, the results indicate that higher levels of FoMO does predict greater engagement in academic misconduct, alcohol drinking, illegal drug use, and other illegal behaviors. Living situation, socioeconomic status, and gender, had several main effects of their own across these behaviors as well as moderating a few of these relationships with FoMO as predicted. Living situation and gender had main effects of their own in predicting engagement of maladaptive behaviors. This suggests that FoMO exists as an aversive phenomenon regarding affect and leads to concrete consequences for individuals and society.

Specifically, higher FoMO was significantly associated with higher rates of plagiarism (before and during college), cheating (before college), and giving away illegal drugs (in college). Furthermore, there were significant interactions between living situation, FoMO, and giving away prescription drugs in college, and socioeconomic status, FoMO, and stealing in college. The interaction between socioeconomic status, FoMO, and plagiarism in college closely approached significance. Additionally, higher FoMO was significantly associated with higher rates of depressant use, stimulant use, cannabis use, and hallucinogen use. FoMO also predicted earlier age beginning alcohol consumption. Furthermore, there was a significant interaction between living situation, FoMO, and typical weekly alcohol consumption.

Supervised machine learning approaches were successfully implemented to predict class membership across various maladaptive behaviors in college students above a random baseline chance of 50% (RQ1). The order in which we can predict these measures (from best to worst): 1) Academic Misconduct / Illegal Behavior (tie), 2) Drugs, 3) Alcohol. The lower accuracy of prediction for alcohol usage is likely partially due to the ubiquity of alcohol use among college students. Alcohol remains the most used substance within the college setting [36]. Alcohol use is likely both normative and accepted within the college student subculture. Moreover, FoMO, and specifically the aggregate score, carried much more predictive importance than other demographic features (RQ2) and individual FoMO items. The fact that the aggregate FoMO score carried much more predictive importance than other demographic features and especially the individual FoMO items is encouraging from a psychometric perspective. These results further confirm that the multi-item measure is appropriate and necessary to capture the complete underlying construct. We get more information and higher predictive power from the aggregate scores compared to any single FoMO indicator. Additionally, these results provide additional predictive validity evidence for the general FoMO measure as aggregate FoMO scores predicted the focal outcomes better than demographic indicators.

## Part 1. Traditional statistical modeling

As predicted by self-determination theory and social comparisons theory, FoMO was shown to play a significant role in influencing higher engagement in various maladaptive behaviors by college students. Specifically, engagement in increased academic misconduct may be due to FoMO's fit within the Conservation of Resources [18] and Social Comparison Theories [19]. A desire to achieve higher grades and the potential future opportunities (i.e., graduate school, a job) that comes because of higher grades may explain willingness to cheat and plagiarism. With regards to higher levels of FoMO predicting substance use, both alcohol and illegal drugs, the relationship might be due to a desire to "fit in" with peers, especially when not engaging in these behaviors may exclude them from parties or other social gatherings. A similar desire to not be removed from social groups can explain the pressure college students with elevated FoMO might feel that leads to engagement in illegal behaviors.

While this study did not directly investigate the mechanism involved in these newly found relationships, it provides a foundation upon which further studies can proceed. Future studies investigating FoMO and these maladaptive behaviors in college students would probe into measurements and manipulation of the key aspects involved in the potential mechanisms of COR, SDT, and SCT. It is likely that it will not just be one, but a combination of several theoretical models behind the relationship of FoMO and maladaptive behaviors.

## Part 2. Machine learning

The results demonstrate that machine learning approaches serve as a powerful tool for carrying out predictive analysis as it relates to the relationship between FoMO and maladaptive behavior. When considering accuracy as a metric, the main conclusion from the results shown in Tables 3 and 4 is that models with a reduced number of features are at least as good as those with a larger number of features. This was observed in two scenarios. First, the models that incorporated a dimensionality reduction technique (RFE or PCA) resulted in improvement in model performance. In some cases, the gain was of a substantial amount. This suggests that those other features that weren't selected may be acting as noise, masking the real signal between input and output. Future measurement can have a reduced number of measures, making data collection more efficient, less demanding on resources, and also more convenient for both the subject and researcher. For example, if the goal was to screen for those college students at-risk accurately and efficiently for problematic drug behaviors, a brief 11 item questionnaire (FoMO measure consisting of ten items and living situation) could be deployed in a matter of seconds and yields a 78% accuracy rate. Second, when we consider the difference in performance between models that incorporated all FoMO items (i.e. individual), and those that used the aggregate FoMO measure as an input feature instead, we can conclude that the observation holds. Using individual FoMO features does not offer an advantage over using the aggregate measure. Results from feature importance highlight the outsize contribution of the aggregate FoMO measure to the models across all domains of behavior. The importance scores of the individual FoMO items show that items carry different predictive weights relative to the four dependent variables. For instance, while FoMO5 ("It is important that I understand my friends "in jokes") has an importance score of 0.28 for drug offense/use, its importance score drops to 0.04 for illegal behavior.

## Practical applications

Although further work is required, the present results already lend themselves to useful application by university and college counselors, especially those focused on assisting new or first-year students transitioning into university for the first time. We found that aggregate FoMO

scores predicted several behaviors likely to disrupt a student's academic career. Counselors working with potentially at-risk students could provide a brief FoMO assessment as it is only a ten-question survey to better understand what risks might be most likely to disrupt that student's college progression or lead to dropping out of the university. With this information in tandem with the tenets of self-determination theory counselors might focus on healthier methods of fulfilling innate needs for social relatedness, competence, and autonomy. Additionally, as higher FoMO students likely engage in more frequent social comparison processes, counselors identifying high FoMO students might seek to redirect those social comparison processes or disrupt them to potentially disrupt future maladaptive behavior. However, that notion requires future work confirming that social comparisons mediate this relationship. Regarding clinical application, this approach has potential for early identification of persons within the at-risk population (i.e., high FoMO). Early identification provides for more systematic and comprehensive research in this area, as well as eventual delineation of treatment options. Moreover, early assessment and detection allows for better understanding of pathogenesis, development of prevention techniques, and prediction of treatment response [37].

From a psychometric perspective, our results might suggest additional avenues by which researchers can gather predictive validity evidence concerning new measure creation. Traditionally, predictive validity evidence gathering involves capturing predictor information at one time point and then capturing outcome information later. If the predictor variable explains unique variance, especially above other known predictors, this is accepted as evidence toward establishing predictive validity. However, a machine learning approach achieves a similar objective using cross-sectional data and advanced classification algorithms. Applying such an approach to future validation attempts provides an additional source of strong information regarding a measure's ability to predict a given outcome. Additionally, machine learning allows us to examine the unique influence of each individual indicator of the focal construct to confirm whether the aggregate score holds the most predictive power relative to any individual item. Future work should consider how such an approach might also be used to reduce the number of items, based on predictive value, for a streamlined measure with the highest predictive potential.

## Limitations and future directions

As with any study, this work had some limitations which should be noted when interpreting the present findings. Due to logistical and resource constraints, the relationships between FoMO and maladaptive behaviors were examined through a cross-sectional study design. Although this provides evidence for the hypothesized relationships, future work should focus on assessing causation. Longitudinal work or daily diary studies might be a particularly profitable method of gathering data. Individuals considering such work might examine social comparison orientation (or acute social comparisons via a diary study), FoMO-related anxiety experiences, and/or need to belong as potential starting points for possible mediators between FoMO and maladaptive college student behaviors.

Another important note regards Part 2 and the use of machine learning. The results we showed represent a baseline performance in terms of the predictive metrics considered. Given a specific modeling goal and scenario, it would be possible to further optimize the models based on a metric of interest. Our goal, however, was to demonstrate a relationship between FoMO and maladaptive behavior and to expand on that work to determine the predictive power of FoMO regarding those behaviors. Given that we observed significant predictive effects of aggregate FoMO in this research, future work might examine specific scenarios

where it would be meaningful to consider further model optimizations or a broader range of machine learning algorithms.

## Conclusion

The results of this study indicate that FoMO has a significant inferential and predictive relationship with maladaptive behaviors in college students. Higher levels of trait FoMO predict higher engagement in several domains of maladaptive behaviors in college students. Furthermore, the aggregate FoMO score was shown to carry the most predictive signal when compared to individual FoMO items and other relevant demographics.

Although this study's original aim was to find initial support for or against FoMO's relationship with maladaptive behaviors, there are now many questions regarding this relationship that remain currently unanswered. Future research should address current limitations as well as extending the scope of analyses and model building.

Lastly, as this study does not identify or suggest any interventions to ameliorate negative consequences of FoMO directly or indirectly, we do suggest that increased screening is given to college students that may be at risk of developing or engaging in harmful behaviors.

## Supporting information

**S1 Fig. Feature importances across all four maladaptive behavior domains.** Mean values of feature importance scores obtained from machine learning models for both modeling scenarios considered, aggregate and individual. The aggregate case includes the metric FoMO Mean as a predictor, whereas the individual scenario uses the 10 FoMO items denoted FoMO 1 to FoMO 10 (but not FoMO Mean). In the aggregate case, FoMO Mean produces the highest importance scores among the predictors across all behavior domains. In the individual case, the scores of the FoMO items vary substantially across behavior domains.
(TIF)

**S1 File. Full regression results.** This file reports the full results for all HLR models for all maladaptive behaviors tested.
(DOCX)

**S2 File. Additional machine learning information.**
(DOCX)

**S3 File. Decision tree output for academic misconduct, alcohol use, and illegal behaviors.** Decision trees for the classification of academic misconduct, alcohol, and illegal behavior based on the FoMO aggregate scenario. Starting at the root node, an example is evaluated in a sequential manner down the tree based on the conditions in the decision nodes. A classification is made according to the end node reached (blue denotes a positive prediction and light orange a negative prediction).
(DOCX)

## Author Contributions

**Conceptualization:** Paul C. McKee, Kenneth S. Walters.

**Data curation:** Paul C. McKee, Kenneth S. Walters.

**Formal analysis:** Paul C. McKee, Christopher J. Budnick, Imad Antonios.

**Funding acquisition:** Paul C. McKee, Christopher J. Budnick.

**Supervision:** Kenneth S. Walters, Imad Antonios.

**Visualization:** Paul C. McKee, Imad Antonios.

**Writing – original draft:** Paul C. McKee, Christopher J. Budnick, Imad Antonios.

**Writing – review & editing:** Paul C. McKee, Christopher J. Budnick, Kenneth S. Walters, Imad Antonios.

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
