## [Decision Letter · Decision Letter 0]

22 Jun 2022

PONE-D-21-36817College Student Fear of Missing Out (FoMO) and Maladaptive Behavior: Traditional Statistical Modeling and Predictive Analysis using Machine LearningPLOS ONE

Dear Dr. McKee,

Thank you for submitting your manuscript to PLOS ONE. After careful consideration, we feel that it has merit but does not fully meet PLOS ONE’s publication criteria as it currently stands. Therefore, we invite you to submit a revised version of the manuscript that addresses the points raised during the review process.

Please see detailed comments from the reviewers below. The reviewers have raised a number of concerns. They request improvements to the reporting of methodological aspects of the study, for example, how the 6 stated hypotheses align with and/or are integrated into the research questions. Can you please carefully revise the manuscript to address all comments raised?

We look forward to receiving your revised manuscript.

Kind regards,

Katrien Janin, PhD

Staff Editor

PLOS ONE

**Journal requirements:**

Reviewers' comments:

Reviewer's Responses to Questions

**Comments to the Author**

1. Is the manuscript technically sound, and do the data support the conclusions?

Reviewer #1: Yes

Reviewer #2: Partly

Reviewer #3: Yes

2. Has the statistical analysis been performed appropriately and rigorously? 

Reviewer #1: Yes

Reviewer #2: Yes

Reviewer #3: Yes

3. Have the authors made all data underlying the findings in their manuscript fully available?

Reviewer #1: Yes

Reviewer #2: No

Reviewer #3: Yes

4. Is the manuscript presented in an intelligible fashion and written in standard English?

Reviewer #1: Yes

Reviewer #2: Yes

Reviewer #3: Yes

5. Review Comments to the Author

Reviewer #1: Overall, according to the reviewing guidelines offered by PLOS ONE, I find this manuscript includes a fair treatment of previous literature in this area, well articulated hypotheses, valid and appropriate data, adequate modeling details, thorough reporting, and thoughtful conclusions.

I applaud the authors for so clearly categorizing and articulating their individual hypotheses. This is excellent analysis and reporting practice and limits so-called “fishing expeditions” when such hypotheses are planned prior to looking at the data and contribute to reproducibility in science. The authors also report all regression statistics, estimates, and p-values, which further contributes to transparency and reproducibility.

The authors provide a clear and concise explanation of the differences between statistical inference and machine learning prediction. Furthermore, the descriptions of test/training sets and k-fold cross-validation were concise and useful and will assist a reader unfamiliar with these concepts. Later in the manuscript, the authors also describe the dual function of some machine learning approaches in providing both feature selection/importance as well as predicted values.

Recommendations:

1. The hypotheses are dropped into the text without introduction. I recommend including a sentence somewhere in the beginning of Part 1 to the effect of, “Six total hypotheses were tested regarding the relationship between FoMO and academic misconduct, substance use, and illegal behaviors…” to help orient the reader.

2. Recommend removing the term “significant” from the stated hypotheses unless the authors wish to outline the parameters under which the association coefficients for each variable will be deemed “clinically significant.” Otherwise, “significant” here is assumed to pertain to the statistical test and this is implied in the hypothesis testing itself.

3. The stated hypotheses specify a direction of effect (“positively”) and a ranking of effect (“strongest”) and neither of these were necessarily directly tested. The hypotheses were tested in the regression context that utilizes two-sided tests, as is usually done, and so I recommend rephrasing the hypotheses to reflect this. Generally, this is backwards – adjusting the hypothesis to fit the test, but in this case, it seems as though this is a point of clarification and specificity of language, not of post-hoc “adjustment.” I recommend altering the working to something along the lines of, “FoMO will be associated with illegal behavior” and “Covariates X, Y, and Z will moderate the above association” to avoid implying one-directional tests that are not actually conducted.

4. For all 3 categories of tests, the second hypothesis includes a sex association, but while the cited literature supports the hypotheses regarding socioeconomic status and living situation, I didn’t note any literature supporting this hypothesis for male sex.

5. It is not immediately clear how the 6 stated hypotheses align with and/or are integrated into the research questions listed on page 6, lines 215-218.

6. The “Part 1” and “Part 2” headings in the 2.2 and 2.3 sections are confusing, perhaps they are misplaced?

7. Recommend rephrasing Page 6, lines 211-212 from, “This work expands our understanding of college student FoMO while contributing to the recent shift toward utilizing multiple statistical approaches” to something along the lines of, “This work leverages multiple complimentary statistical and machine learning approaches to expand our understanding of college student FoMO,” as implementing both inferential and predictive methods together is arguably not a recent development.

8. The justification for categorizing the maladaptive behavior measures on page 8 is difficult to follow. The statements, “Furthermore, the current approach was data driven. Hence, it was preferable to use the more efficient binary classifications so long as a dimensional approach was not more accurate” seems to imply that the authors tested multiple approaches (binary and dimensional) but this doesn’t seem to have been done? I might recommend simply removing those statements altogether and simply stating that as a initial analysis, considering that binary classifications are typically those clinically utilized, a binary classification approach was adopted. Then I recommend adding some sentences in the limitations/future work section to suggest that exploring the fully dimensionality of the behavioral measurements may be of future interest.

9. The results on pages 14-16 are difficult to parse in the text. I recommend moving the F statistics and degrees of freedom to the table and removing those and the beta estimates and p-values from the text – so long as these values are reported in the table, they do not need also to be repeated in the text. It may also be useful to revisit editing this section for conciseness. Furthermore, there is no reference to this table in the text.

10. Recommend reiterating for the reader in the paragraph beginning on page 18 (line numbers not available here) that the “Aggregate” FoMO value is the mean score while the “Individual” is the sum of items.

11. In contrast to item 3, noted above, the results for the machine learning section are sparse, with only tables and figures and very little text explaining these. I recommend adding some sentences summarizing the findings in the table. Some of the performance metrics information from the discussion on page 23 in particular might be better suited moved to the results section.

12. The in-text references to tables on page 18 are misnumbered

13. Recommend adding an overall caption for the tables in the supplement

14. One of the supplemental tables is blank

15. I highly recommend that the authors consider making the analytical code for the machine learning approaches publicly available via GitHub or some other platform. Given that machine learning is not broadly, openly acceptable as a standard analysis approach by everyone in the social, behavioral, and biological sciences, making code openly available contributes to transparency and reproducibility in science and builds trust among our collaborators.

Reviewer #2: Summary

This paper briefly presents the influence of the Fear of Missing Out (FoMO) on academic misconduct, drug use and illegal behavior as part of the self-determination theory (SCT). This results in six hypothesis that FoMO, taking sociodemographic variables into account, influences this behavior in college students.

A second goal of the work is to investigate the extent to which machine learning brings further advantages in comparison or in combination with classical statistical methods.

The results confirm the assumptions that FoMO alone as the strongest predictor and partly in connection with the SES and living conditions (alone, on campus, with parents) predict academic misconduct, alcohol and drug use and illegal behavior (petty crime).

General remarks

The biggest weakness of the paper is the lack of descriptive presentation of the results in Part 2 (Machine Learning), where only sparsely annotated tables and figures are presented. What is needed here is a detailed description of the machine learning results. If necessary, there should also be individual explanations of what the values mean, so that readers with little or no knowledge of machine learning can understand the results.

Strong points are the presentation and justification of the machine learning methods used, which some readers may not yet be familiar with, and the presentation of the theoretical background for the content of the study, although it is somewhat brief.

The results of the classical analyses are described correctly and in detail and presented in tables.

Since some of the methods in Part 2 with machine learning are also used in classical statistics, this should be clarified in the methods section. How does machine learning differ from classical statistics? Where do they overlap? The current version gives the impression that e.g. PCA belongs to machine learning. But there is no such clear distinction.

Abstract

FoMO; Fear of Missing Out not only abrevation (even if written in titel) and evtl. short explanation also in abstract

PCA and logisic regression are part also part of the 'classical' statistics in psychology and not genuine ML

What extcatly are " additional insights that would not be possible through statistical modeling approaches"?

Keywords

"drug usepredictive analysis" -> "drug use, predictive analysis"

+ academic misconduct, illegal behavior, alcohol use

Major Issues

Introduction

I recommend presenting the relationship between FoMO and maladaptive behaviour more stringently in the theory section. Even though there are direct and indirect relationships between anxiety, depressiveness and academic misconduct, I would leave out internalising problems here or argue more precisely if this is important in the context of SDT. Also, the link to increased Facebook use during classroom lectures does not seem very relevant to me. If anything, I would report more generally on the use of social networks in school in connection with FoMo (e.g. a meta analysis), or leave this out.

A thought you might consider: What is the relationship of FoMO to procrastination? It seems plausible to me that FoMO leads to procrastination.

see points under General Remarks

Methods

249 Do you have a reference for this questionnaire?

Chapter 2.3 Data Analysis

"A series of hierarchical regression analyses were conducted to test the association between trait level FoMO and engagement in a broad range of maladaptive behaviors during college. For each dependent variable of interest, there were three separate regression models run."

If you a regression analysis for each dependent variable (academic misconduct, alcohol use, drug use, illegal behavior) I recommend a correction of the significance level (e.g. bonferoni correction) or a multivariat regression analysis.

p 14 "Hypothessi Testing"

As far as I understood you did different hierarchical regression analyses.

Did you correct the significance level?

Why didn't you use a multiple hierarchical regression analyses integrating all or at least several independent variables?

307 2.3 Analysis: This sections repeats most information of "Hypotheses testing" in section 2.3 on line 256.

323 "There were no missing item-level data as the dataset was screened and cleaned prior to Part 1 of this study."

-> I would mention this already in the sections of part 1, as I suppose you did the data cleaning for all analyses.

As there are no missing data after data cleaning, how many subjects were excluded? Could these missing data have an influence on the results.

329 "logistic regression"

There should be a short discussion about what is machine learning and what belongs to classical analyses in psychology. What's about overlapping methods?

logistic regression and PCA are often used in classical psychological research.

I recommend to discuss this already earlier in the paper (e.g. a new section in the introduction); earlier than line 344

p. 18 "Part 2"

you indicate table 1 and 2 and 3 instead of Table 3 and 4 and 5

And I miss a description of the tables.

What is shown there. Which values are important.

Table 5 What do the numbers mean?

In the classical statistics section you descriped in details the results, in this section there is nearly no text to explain the tables and figures.

I miss a description of the figures 1, 2, and 3.

What is shown in the figures and what is the main information in the figures.

Disscussion

p. 21 General

I would mention the confirmation of the hypotheses within the results section.

The text here is ok, but as the hypothses H2, H4, and H6 are more complicate to describe I would omit here to mention the hypothses.

you mention the hypotheses H2, H3 and H4 instead of H3a, H3b and H5

p. 21

At the end of this section (p. 22) I miss a short discussion of the added value of combining the two evaluation methods (classical analysis, machine learning). Are there contradictions that are not to be expected? Do the methods complement each other? Does machine learning improve the classical methods or could it be replaced by machine learning? Or is machine learning not necessary to arrive at the results?

Although the results of Part 2 (Machine Learning) are discussed on their own on p. 23f, in my opinion they are not put into context with the results of the classical method. This is done a little at the beginning of p. 25.

p. 25

" Additionally, machine learning allows us to examine the unique influence of each individual indicator of the focal construct to confirm whether the aggregate score holds the most predictive power relative to any individual item"

-> But, the the hierarchical regression analysis does show this as well. What's the gain of machine learning here?

"a cross-sectional study design"

This is also true for part 2.

p 25 Limitations and Future Directions

There is "never enough data" for machine learning. Therefore, it is certainly a weakness that relatively little data is available.

Minor Issues

Introduction

117 unnecessary comma "maladaptive, behaviors"

121, 165 "FOMO" (big O)

158 H3: drug use; Which drug? I would list the analyzed drugs

168ff "Although research is limited, some findings suggest that high FoMO individuals are more likely to engage in low-level illegal behavior such as driving while using a cell phone" -> I miss here references to "some findings suggest"

178 I miss here a logic for titeling.

Part 1 [no subtitle] is about content with hypothesis for inference statistics

Part 2: "Statistical Modeling Approaches" is about machine learning techniques

207 evtl. missing reference "(2020)"

Methods

279ff "While clinical diagnosis is slowly moving toward more dimensional approaches, diagnostic classification remains the long-established norm, especially in clinical practice (Woo & Keatinge, 2016)."

That's an argument. But your instruments are not constructed to make diagnoses. So, there is an annalogie to clinical diagnostics, but here it seems that you actually do clinical diagnostic classification.

307 Chapter number 2.3 already used on line 256

331 "review (Kotsiantis, 2007)" -> "review Kotsiantis (2007) for"

Results

Table 1: The Cronbach's Alpha in line 1 and 2 of the table are confusing; they seem to be correlations with the variables itself. I would report these values in the text and not in this table.

after line 409 the numbering stops

Discussion

"General" I would change this subtitle to "Summary"

General

I would use the word gender instead of the word sex.

Reviewer #3: The paper is highly commendable. The topic is very timely and the analysis using the different data analytical tools produced impressive findings that could help scientist and experts in the field of behavioral sciences understand what FOMO is. However, the abstract should be improved. It only focused on the data analytical tools instead of the results and conclusions derived from the study which could help the readers understand what FOMO is and its relationship with some maladaptive behaviors. Authors may consider addressing this issue. Also, proper documentation of in-text citations should be observed. A large majority of the authors and works cited in the text are not listed in the references. This may derail the brevity of the study and its findings. All in all, the paper is highly acceptable.

6. PLOS authors have the option to publish the peer review history of their article (what does this mean?). If published, this will include your full peer review and any attached files.

Reviewer #1: No

Reviewer #2: No

Reviewer #3: **Yes: **Prof. Gino A. Cabrera

---

## [Author Response · Author response to Decision Letter 0]

26 Jul 2022

Reviewer #1: Overall, according to the reviewing guidelines offered by PLOS ONE, I find this manuscript includes a fair treatment of previous literature in this area, well articulated hypotheses, valid and appropriate data, adequate modeling details, thorough reporting, and thoughtful conclusions.

I applaud the authors for so clearly categorizing and articulating their individual hypotheses. This is excellent analysis and reporting practice and limits so-called “fishing expeditions” when such hypotheses are planned prior to looking at the data and contribute to reproducibility in science. The authors also report all regression statistics, estimates, and p-values, which further contributes to transparency and reproducibility.

The authors provide a clear and concise explanation of the differences between statistical inference and machine learning prediction. Furthermore, the descriptions of test/training sets and k-fold cross-validation were concise and useful and will assist a reader unfamiliar with these concepts. Later in the manuscript, the authors also describe the dual function of some machine learning approaches in providing both feature selection/importance as well as predicted values.

We thank you for your time reviewing this as well as the suggestions to make this work even stronger.

Recommendations:

1. The hypotheses are dropped into the text without introduction. I recommend including a sentence somewhere in the beginning of Part 1 to the effect of, “Six total hypotheses were tested regarding the relationship between FoMO and academic misconduct, substance use, and illegal behaviors…” to help orient the reader.

We added a sentence to the end of the first and second paragraphs of the introduction to help orient the readers to our hypotheses and research questions, respectively. 

“Six hypotheses tested the relationship between FoMO, with relevant moderating demographic variables, and academic misconduct, drug use, alcohol use, and illegal behaviors.”

“To this end we asked two research questions examining if FoMO can predict behavior above chance, and if so, how much weight does it carry compared to other variables.”

2. Recommend removing the term “significant” from the stated hypotheses unless the authors wish to outline the parameters under which the association coefficients for each variable will be deemed “clinically significant.” Otherwise, “significant” here is assumed to pertain to the statistical test and this is implied in the hypothesis testing itself.

We have removed the term ‘significant’ as requested - thank you for pointing this out.

H1: Higher FoMO levels will be associated with academic misconduct. 

H3: FoMO will be associated with drinking behavior (a) and drug use (b).

H5: FoMO will be associated with illegal behavior.

3. The stated hypotheses specify a direction of effect (“positively”) and a ranking of effect (“strongest”) and neither of these were necessarily directly tested. The hypotheses were tested in the regression context that utilizes two-sided tests, as is usually done, and so I recommend rephrasing the hypotheses to reflect this. Generally, this is backwards – adjusting the hypothesis to fit the test, but in this case, it seems as though this is a point of clarification and specificity of language, not of post-hoc “adjustment.” I recommend altering the working to something along the lines of, “FoMO will be associated with illegal behavior” and “Covariates X, Y, and Z will moderate the above association” to avoid implying one-directional tests that are not actually conducted.

Thank you for this comment - we have rephrased our hypotheses accordingly.

H1: Higher FoMO levels will be associated with academic misconduct. 

H2: Living situation (a), SES (b), and sex (c) will moderate the above relationship. 

H3: FoMO will be associated with drinking behavior (a) and drug use (b).

H4: Living situation (a), SES (b), and sex (c) will moderate the above relationship. 

 H5: FoMO will be associated with illegal behavior.

H6: Living situation (a), SES (b), and sex (c) will moderate the above relationship.

4. For all 3 categories of tests, the second hypothesis includes a sex association, but while the cited literature supports the hypotheses regarding socioeconomic status and living situation, I didn’t note any literature supporting this hypothesis for male sex.

Thank you for pointing this out. We added relevant literature and citations supporting the hypotheses for sex associations. 

“It has also been found that males generally report higher levels of academic misconduct compared to females (Whitley et al, 1999).”

“Illicit drug, nicotine, and alcohol use is much more prevalent in men than with women, although the relationship with alcohol seems to disappear among adolescents (ages 12-17) (Center for Behavioral Health Statistics and Quality, 2016).”

“Moreover, gender is one of the strongest predictors of delinquency and violent criminal behavior with males being perpetrators at much higher rates than females (Mears et al., 1998, Heidensogn, 1997). “

5. It is not immediately clear how the 6 stated hypotheses align with and/or are integrated into the research questions listed on page 6, lines 215-218.

We have reworded this section to make it more clear how these hypotheses and research questions are related.

“Therefore, in Part 1 we identify relationships via traditional methods (i.e., hierarchical linear regression) and in Part 2 we use machine learning to address two research questions that build off those previous hypotheses:

RQ1: If FoMO is found to have relationships with different maladaptive behaviors, can machine learning algorithms predict those behaviors in college students beyond random chance?

RQ2: If FoMO is found to have relationships with different maladaptive behaviors and machine learning algorithms can predict those behaviors in college students beyond random chance, how much predictive weight will FoMO carry compared to other demographic features?”

6. The “Part 1” and “Part 2” headings in the 2.2 and 2.3 sections are confusing, perhaps they are misplaced?

Thank you for pointing this out. All headings have been fixed.

7. Recommend rephrasing Page 6, lines 211-212 from, “This work expands our understanding of college student FoMO while contributing to the recent shift toward utilizing multiple statistical approaches” to something along the lines of, “This work leverages multiple complimentary statistical and machine learning approaches to expand our understanding of college student FoMO,” as implementing both inferential and predictive methods together is arguably not a recent development.

We have revised this sentence as suggested.

“This work expands our understanding of college student FoMO by leveraging complementary and convergent statistical and machine learning approaches.”

8. The justification for categorizing the maladaptive behavior measures on page 8 is difficult to follow. The statements, “Furthermore, the current approach was data driven. Hence, it was preferable to use the more efficient binary classifications so long as a dimensional approach was not more accurate” seems to imply that the authors tested multiple approaches (binary and dimensional) but this doesn’t seem to have been done? I might recommend simply removing those statements altogether and simply stating that as a initial analysis, considering that binary classifications are typically those clinically utilized, a binary classification approach was adopted. Then I recommend adding some sentences in the limitations/future work section to suggest that exploring the fully dimensionality of the behavioral measurements may be of future interest.

Thank you for the suggestion. We have removed and replaced the wording as suggested. We did have a sentence already after that statement that talks about future research of exploring higher dimensionality.

 “Hence, as an initial analysis it was preferable to use the binary classifications that are typically clinically used. Future research can investigate more nuanced and specific expanded classification problems (e.g., nonuser/experimenter/heavy drug user).”

9. The results on pages 14-16 are difficult to parse in the text. I recommend moving the F statistics and degrees of freedom to the table and removing those and the beta estimates and p-values from the text – so long as these values are reported in the table, they do not need also to be repeated in the text. It may also be useful to revisit editing this section for conciseness. Furthermore, there is no reference to this table in the text.

At your suggestion we have removed the statistics from the text and left just the words. Additionally, we added two brief sentences are the end of the first section of results (Academic Misconduct) directing readers to both Table 2 to see the summary of found relationships as well as the supplemental materials to see the full results.

“See Table 2 for a summary of found relationships. See supplemental materials for full results.”

10. Recommend reiterating for the reader in the paragraph beginning on page 18 (line numbers not available here) that the “Aggregate” FoMO value is the mean score while the “Individual” is the sum of items.

We have added a sentence reminding readers of this as suggested.

“Please note that “Aggregate” refers to using just the mean score across all ten FoMO items as a single predictor while “Individual” refers to using the score of each of the ten items as separate predictors.”

11. In contrast to item 3, noted above, the results for the machine learning section are sparse, with only tables and figures and very little text explaining these. I recommend adding some sentences summarizing the findings in the table. Some of the performance metrics information from the discussion on page 23 in particular might be better suited moved to the results section.

We have substantially expanded the description of the machine learning results (tables and figures) and included explanations of how to interpret the various metrics presented. We have also shifted some text from the discussion section as suggested. 

12. The in-text references to tables on page 18 are misnumbered

Thank you for pointing this out - it has been fixed.

 “In Table 3 and 4 below, we show the results of applying the classifiers to predict the four variables of interest. For each of the measures, we show the achieved accuracy, F1-score, and ROC AUC (denoted by ROC in the table header) using the two modeling scenarios described earlier, denoted as “Aggregate” and “Individual” in the tables. Following those tables is a figure comparing accuracy for all models across “Aggregate” and “Individual” approaches. Then, we show decision tree output for each of the four domains of behavior. In Table 5, we show the average feature importance across all models for each of the variables considered in the aggregate and individual cases.”

13. Recommend adding an overall caption for the tables in the supplement

Thank you for pointing this out - we have addressed it.

14. One of the supplemental tables is blank

Thank you for pointing this out - we have addressed it.

15. I highly recommend that the authors consider making the analytical code for the machine learning approaches publicly available via GitHub or some other platform. Given that machine learning is not broadly, openly acceptable as a standard analysis approach by everyone in the social, behavioral, and biological sciences, making code openly available contributes to transparency and reproducibility in science and builds trust among our collaborators.

The analytical code for the machine learning as well as the data itself will be made publicly available at OSF at the following link at the time of publication.

https://osf.io/r7xyn/?view_only=8191203963dd46ae87996116102cf305

Reviewer #2: Summary

This paper briefly presents the influence of the Fear of Missing Out (FoMO) on academic misconduct, drug use and illegal behavior as part of the self-determination theory (SCT). This results in six hypothesis that FoMO, taking sociodemographic variables into account, influences this behavior in college students.

A second goal of the work is to investigate the extent to which machine learning brings further advantages in comparison or in combination with classical statistical methods.

The results confirm the assumptions that FoMO alone as the strongest predictor and partly in connection with the SES and living conditions (alone, on campus, with parents) predict academic misconduct, alcohol and drug use and illegal behavior (petty crime).

General remarks

The biggest weakness of the paper is the lack of descriptive presentation of the results in Part 2 (Machine Learning), where only sparsely annotated tables and figures are presented. What is needed here is a detailed description of the machine learning results. If necessary, there should also be individual explanations of what the values mean, so that readers with little or no knowledge of machine learning can understand the results.

We have substantially expanded the description of the machine learning results (tables and figures) and included explanations of how to interpret the various metrics presented. We have also shifted some text from the discussion section to results for improved readability. 

Strong points are the presentation and justification of the machine learning methods used, which some readers may not yet be familiar with, and the presentation of the theoretical background for the content of the study, although it is somewhat brief.

Thank you.

The results of the classical analyses are described correctly and in detail and presented in tables.

Thank you.

Since some of the methods in Part 2 with machine learning are also used in classical statistics, this should be clarified in the methods section. How does machine learning differ from classical statistics? Where do they overlap? The current version gives the impression that e.g. PCA belongs to machine learning. But there is no such clear distinction.

We have added the following text in the introduction to highlight the differences between the two approaches used in the paper:

“The differences between the two approaches employed in our paper have been a subject of some debate, so we include some brief comments to highlight these differences. For a more detailed treatment, the reader is directed to Bzdok, Altman & Krzywinski 2018. While machine learning is built on a statistical framework and often includes methods that are employed in statistical modeling, its methods also draw on fields such as optimization, matrix algebra, and computational techniques in computer science. The primary difference between the two approaches is in how they are applied to a problem and what goals they achieve. Statistical inference is concerned with proving the relationship between data and the dependent variable to a degree of statistical significance, while the primary aim of machine learning is to obtain the best performing model to make repeatable predictions. This is achieved by using a test set of data as described earlier to infer how the algorithm would be expected to perform on future observations. When prediction is the goal, a large number of models are evaluated and the one with the best performance according to a metric of interest is deployed.”

Abstract

FoMO; Fear of Missing Out not only abrevation (even if written in titel) and evtl. short explanation also in abstract

Thank you. We have added this as requested.

“This paper reports a two-part study examining the relationship between fear of missing out (FoMO) and maladaptive behaviors in college students.”

PCA and logisic regression are part also part of the 'classical' statistics in psychology and not genuine ML

What extcatly are " additional insights that would not be possible through statistical modeling approaches"?

We made this as clear as possible given the limited space in the abstract. 

“This study demonstrated FoMO’s relationships with these behaviors as well as how machine learning can provide additional predictive insights that would not be possible through inferential statistical modeling approaches typically employed in psychology, and more broadly, the social sciences.” 

Keywords

"drug usepredictive analysis" -> "drug use, predictive analysis"

+ academic misconduct, illegal behavior, alcohol use

Thank you. We have updated the key words.

Keywords - FoMO, machine learning, college students, alcohol use, drug use, academic misconduct, illegal behavior, predictive analysis.

Major Issues

Introduction

I recommend presenting the relationship between FoMO and maladaptive behaviour more stringently in the theory section. Even though there are direct and indirect relationships between anxiety, depressiveness and academic misconduct, I would leave out internalising problems here or argue more precisely if this is important in the context of SDT. Also, the link to increased Facebook use during classroom lectures does not seem very relevant to me. If anything, I would report more generally on the use of social networks in school in connection with FoMo (e.g. a meta analysis), or leave this out.

We appreciate this suggestion and have added brief (given space considerations for this outlet) additions to better clarify the positioning of SDT. For examples see:

P.4 “Thus, underperforming students might be more likely to engage in cheating or other academic misconduct to increase their career resources and status when socially comparing themselves to others because underperformance could suggest a threat to competence need fulfillment as SDT suggests. “

P.4 “To reduce FoMO, students might use substances to “fit in” or belong in a peer group to fulfill social relatedness needs.“

P. 5 “Per COR theory, the threat of being left out may be experienced as a threat to one’s status, social relatedness, or reputational resources – needs requiring fulfillment for wellbeing and motivation per SDT. “

As for the link between Facebook use in the class and the present research, we just present that as one example of how fomo can relate to academic incivility. The use of electronic media during a live course for social purposes is considered a form of academic misconduct in the relevant literature.

A thought you might consider: What is the relationship of FoMO to procrastination? It seems plausible to me that FoMO leads to procrastination.

We appreciate this insight, however the current investigation focused on several maladaptive behaviors likely potentially more impactful than procrastination. Although we agree this is an interesting direction for future research we did not collect information regarding procrastination levels and thus cannot assess that with the current data.

see points under General Remarks

Methods

249 Do you have a reference for this questionnaire?

No reference as there was no previously existing measure for conduct problems specifically among college students that suited our needs. So, we created a face valid, straightforward questionnaire to measure the frequency of those behaviors, "since entering college" (not prior). Items were simply asking if students used or engaged in selected drugs/behaviors since entering college.

Chapter 2.3 Data Analysis

"A series of hierarchical regression analyses were conducted to test the association between trait level FoMO and engagement in a broad range of maladaptive behaviors during college. For each dependent variable of interest, there were three separate regression models run."

If you a regression analysis for each dependent variable (academic misconduct, alcohol use, drug use, illegal behavior) I recommend a correction of the significance level (e.g. bonferoni correction) or a multivariat regression analysis.

We appreciate these suggestions, however multivariate analyses would not have achieved the objective of this work. As this is relatively new work connecting FoMO to the examined outcomes while examining moderation effects, whether fomo and focal moderators uniquely influenced each outcome was of particular interest. Given that multivariate analyses actually predict the centroid of the unique combination of all outcomes, such an analysis would not have captured key information required to inform the field and guide future research focused on any single one of the outcomes examined herein.

p 14 "Hypothessi Testing"

As far as I understood you did different hierarchical regression analyses.

Did you correct the significance level?

Given this initial examination of these relationships, we did not correct the significance level. We were interested in identifying any possible relationships among the proposed variables and as such accepted the increased chance of Type I error inherent in multiple testing. Given these initial findings require confirmation in additional samples, these findings thus provide impetus and guidance for future research in this domain. We hope that future researchers will build on this work by isolating key outcomes and further examining potential relationships with FoMO and other important moderators (social comparison orientation, need to belong, etc.). Moreover, the machine learning results confirm our primary findings and thus provide additional support for the results presented in Part I.

Why didn't you use a multiple hierarchical regression analyses integrating all or at least several independent variables?

The focus of this investigation was to better understand how fomo interacted with individual demographic variables on the focal outcomes. Adding additional moderating variables to the model would have expanded that model to include at least one three-way interaction. Thus this approach would not have aligned well with the test of the research questions and may have presented power issues that could have undermined the results. Moreover, including additional variables would have altered the interpretation of the regression models such that any relationship would need to be interpreted as existing at average levels of all moderators in the model. Therefore, we chose to focus on individual analyses to better identify the individual relationships in question.

307 2.3 Analysis: This sections repeats most information of "Hypotheses testing" in section 2.3 on line 256.

Thank you for pointing this out - it has been addressed.

“All statistical analyses were run by IBM SPSS Version 26.0 statistical software package. A series of hierarchical regression analyses were conducted to test the association between trait level FoMO and engagement in a broad range of maladaptive behaviors during college. For each dependent variable of interest, there were three separate regression models run. On Step 1, an alternating demographic variable of interest (gender, socioeconomic status, living situation) and FoMO were entered. We dummy-coded living situations (living with parents = 0) for analysis. To test for a potential interaction of trait FoMO and demographic on the criterion variables, FoMO X Demographic was entered at Step 2 of the regression models. Note, not all possible outcome variables included in the measures (e.g., all illegal behaviors, all drug classes) were analyzed as part of the hypothesis testing. Nonetheless, we included them in the correlation tables so that future research may use any potential information as a foundation for hypothesis or exploratory testing. Given the number of tests we report, we also have truncated several of the results reports to the most pertinent statistical information. Full model results for all statistical tests can be viewed in the online supplemental material.”

323 "There were no missing item-level data as the dataset was screened and cleaned prior to Part 1 of this study."

This sentence has been deleted and embedded in the method revision on p6. (see next comment for full text).

-> I would mention this already in the sections of part 1, as I suppose you did the data cleaning for all analyses.

We appreciate the reviewer pointing this out. We have updated p. 6 of the manuscript to contain additional information about missing data. That revised section now reads: “Four hundred and ninety undergraduate participants from a Northeastern university completed our cross-sectional survey. However, we excluded 18 participants that were not in the targeted age range (i.e.,18-24 years), leaving a final analyzed sample of n = 472 participants with no missing item-level data (Mage = 19.06, SDage = 1.17; 52% white, 23% black, 4% Asian, .2% Pacific Islander/Alaskan Native; 28% male).”

As there are no missing data after data cleaning, how many subjects were excluded? Could these missing data have an influence on the results.

We appreciate the reviewer pointing this out. We have updated p. 6 of the manuscript to contain additional information about missing data. That revised section now reads: “Four hundred and ninety undergraduate participants from a Northeastern university completed our cross-sectional survey. However, we excluded 18 participants that were not in the targeted age range (i.e.,18-24 years), leaving a final analyzed sample of n = 472 participants with no missing item-level data (Mage = 19.06, SDage = 1.17; 52% white, 23% black, 4% Asian, .2% Pacific Islander/Alaskan Native; 28% male).”

329 "logistic regression"

There should be a short discussion about what is machine learning and what belongs to classical analyses in psychology. What's about overlapping methods?

logistic regression and PCA are often used in classical psychological research.

I recommend to discuss this already earlier in the paper (e.g. a new section in the introduction); earlier than line 344

We have added the following text in the introduction to highlight the differences between the two approaches:

“The differences between the two approaches employed in our paper have been a subject of some debate, so we include some brief comments to highlight these differences. For a more detailed treatment, the reader is directed to Bzdok, Altman & Krzywinski 2018. While machine learning is built on a statistical framework and often includes methods that are employed in statistical modeling, its methods also draw on fields such as optimization, matrix algebra, and computational techniques in computer science. The primary difference between the two approaches is in how they are applied to a problem and what goals they achieve. Statistical inference is concerned with proving the relationship between data and the dependent variable to a degree of statistical significance, while the primary aim of machine learning is to obtain the best performing model to make repeatable predictions. This is achieved by using a test set of data as described earlier to infer how the algorithm would be expected to perform on future observations. When prediction is the goal, a large number of models are evaluated and the one with the best performance according to a metric of interest is deployed.” 

p. 18 "Part 2"

you indicate table 1 and 2 and 3 instead of Table 3 and 4 and 5

Thank you for drawing our attention to this - it has been fixed.

And I miss a description of the tables.

Thank you for drawing our attention to this - it has been fixed.

Table 4

Performance metrics (accuracy, F1-score and ROC AUC) obtained from each of the machine learning models across behavior domains using the individual FoMO items as predictors. Consistent with the aggregate FoMO scenario, those models that combined a dimensionality reduction techniques with random forests (RFE + RF and PCA + RF) achieved the highest accuracy for all behavior domains with the exception of alcohol consumption. Using the individual scores does not appear to improve the model predictions compared to the aggregate scenario. 

Table 5

Mean feature importance scores obtained from machine learning models considered across behavior domains. An importance score measures the individual contribution of the feature to the classifier. The higher the score, the higher the contribution to the model. The aggregate FoMO metric (denoted 'FoMO Mean' in the table) has a substantially higher importance score than all other predictors across all behavior domains. When considering the individual scenario, importance scores for FoMO items vary substantially across behavior domains. For instance, 'FoMO 8' has an importance score of 0.24 with respect to academic misconduct but only 0.03 for illegal behavior. 

What is shown there. Which values are important.

Table 5 What do the numbers mean?

Thank you for drawing our attention to this - it has been fixed.

Mean feature importance scores obtained from machine learning models considered across behavior domains. An importance score measures the individual contribution of the feature to the classifier. The higher the score, the higher the contribution to the model. The aggregate FoMO metric (denoted 'FoMO Mean' in the table) has a substantially higher importance score than all other predictors across all behavior domains. When considering the individual scenario, importance scores for FoMO items vary substantially across behavior domains. For instance, 'FoMO 8' has an importance score of 0.24 with respect to academic misconduct but only 0.03 for illegal behavior. 

In the classical statistics section you descriped in details the results, in this section there is nearly no text to explain the tables and figures.

Thank you for pointing this out - we have fixed this.

I miss a description of the figures 1, 2, and 3.

Thank you for pointing this out. We have decided to remove Figure 3 (and place it as supplemental instead) and included descriptive captions for Figures 1 and 2.

Figure 1

Comparison of model accuracy for the aggregate scenario vs. the individual scenario across behavior domains based on values shown in Table 3 and 4. Results are aggregated by machine learning model and scenario, with solid bars representing accuracy values for the aggregate scenario and the bars with patterns showing the results for the individual scenario. 

Figure 2

Decision tree for drug offense/use classification based on the FoMO aggregate scenario. Starting at the root node, an example is evaluated in a sequential manner down the tree based on the conditions in the decision nodes. A classification is made according to the end node reached (blue denotes a positive prediction and light orange a negative prediction). 

What is shown in the figures and what is the main information in the figures.

We have substantially expanded the description of the machine learning results (tables and figures) and included explanations of how to interpret the various metrics presented. We have also shifted some text from the discussion section to results for improved readability. 

Disscussion

p. 21 General

I would mention the confirmation of the hypotheses within the results section.

The text here is ok, but as the hypothses H2, H4, and H6 are more complicate to describe I would omit here to mention the hypothses.

I have added a brief sentence to each section of the results (academic misconduct, alcohol use, drug use, and illegal behaviors, respectively). 

“Together these results provide support for H1 and H2b, although we could not reject the null hypothesis for H2a and H2c.”

“Thus, we were unable to reject the null hypothesis for H2a, H2b, or H3c, but found support for H1.”

“Taken together we found support for H1, but could not reject the null hypothesis for H2a, H2b, or H2c.”

“The results provide support for H1, H2a, and H2b, while we could not reject the null hypothesis for H2c.”

you mention the hypotheses H2, H3 and H4 instead of H3a, H3b and H5

This has been fixed.

p. 21

At the end of this section (p. 22) I miss a short discussion of the added value of combining the two evaluation methods (classical analysis, machine learning). Are there contradictions that are not to be expected? Do the methods complement each other? Does machine learning improve the classical methods or could it be replaced by machine learning? Or is machine learning not necessary to arrive at the results?

We added some text at the end of the introduction to highlight the differences between the two approaches and expanded the machine learning section in results to better explain how to interpret the machine learning models. To summarize, machine learning is valuable in instances where the goal is to build and validate a model that produces the best predictive performance on future observations, whereas the primary aim of classical methods is to infer if the relationship between input variables and the outcome likely exist. Since the aims of the two approaches are different, the choice of which approach(es) to apply should be driven by the research aims. In the context of our study, the practical applications described in the discussion section (i.e. screening test) motivated the need for exploring the effectiveness of machine learning models for predictive analysis. Our machine learning results indicate that models can be expected to perform reasonably well with respect to the metrics considered. 

Although the results of Part 2 (Machine Learning) are discussed on their own on p. 23f, in my opinion they are not put into context with the results of the classical method. This is done a little at the beginning of p. 25.

We added some text at the end of the introduction to highlight the differences between the two approaches and expanded the machine learning section in results to better explain how to interpret the machine learning models. To summarize, machine learning is valuable in instances where the goal is to build and validate a model that produces the best predictive performance on future observations, whereas the primary aim of classical methods is to infer if the relationship between input variables and the outcome likely exist. Since the aims of the two approaches are different, the choice of which approach(es) to apply should be driven by the research aims. In the context of our study, the practical applications described in the discussion section (i.e. screening test) motivated the need for exploring the effectiveness of machine learning models for predictive analysis. Our machine learning results indicate that models can be expected to perform reasonably well with respect to the metrics considered. 

p. 25

" Additionally, machine learning allows us to examine the unique influence of each individual indicator of the focal construct to confirm whether the aggregate score holds the most predictive power relative to any individual item"

-> But, the the hierarchical regression analysis does show this as well. What's the gain of machine learning here?

Although hierarchical regression could test each individual item uniquely, the size of that model would require a sample size that is unrealistic in order to have adequate power to test ten items as unique variables on even a single outcome. Thus machine learning's use of multiple comparison models generated based on the input data overcomes the power issue of too many variables/items in the model given the sample size. Additionally, the information gained from machine learning allows us to build and validate a model that produces the best predictive performance, whereas the information gained from the hierarchical regression analysis is to infer if the relationship between input variables and the outcome likely exists. Since the aims of and information gained from the two approaches are different,it is valuable to have both.

"a cross-sectional study design"

This is also true for part 2.

We have edited this sentence to remove the “Part 1” and make a general statement about the data.

“Due to logistical and resource constraints, the relationships between FoMO and maladaptive behaviors were examined through a cross-sectional study design.”

p 25 Limitations and Future Directions

There is "never enough data" for machine learning. Therefore, it is certainly a weakness that relatively little data is available.

While in general more data can improve the performance of a machine learning model, this is generally observed in neural network models with high-dimensional datasets (i.e. > 100 features). With low-dimensional datasets, the performance of standard machine learning models such as random forests converge relatively quickly, and adding more data does not necessarily improve performance. Since the machine learning results presented in the paper fall under the latter scenario, we feel that we cannot state with much confidence that more data would improve model performance without conducting more detailed analyses (e.g. generating the learning curve for each of the models). Such an analysis is beyond the scope of the paper. Therefore we prefer to exclude any conjecture on sample size as a limitation of the study. 

Minor Issues

Introduction

117 unnecessary comma "maladaptive, behaviors"

This has been fixed.

121, 165 "FOMO" (big O)

This has been fixed.

158 H3: drug use; Which drug? I would list the analyzed drugs

We have included the list of drugs in the Methods/Measures section.

“The drug classes were: marijuana, “powder” cocaine, “crack” cocaine, amphetamines (speed), methamphetamine (Meth), opiates (heroin, etc.), pain medications used for non-medical purposes (Oxycontin, Percocet, etc.), Methadone, barbiturates (downers), tranquilizers (Valium, Xanax, etc.), hallucinogens (LSD, mushrooms, etc.), “club drugs” (ecstasy, ketamine, etc.), inhalants (paint, fumes, etc.), and other non-pain killer prescription medications sued for non-medical purposes (Ritalin, Adderall, etc.).”

168ff "Although research is limited, some findings suggest that high FoMO individuals are more likely to engage in low-level illegal behavior such as driving while using a cell phone" -> I miss here references to "some findings suggest"

Thank you for pointing this missing reference out. We have included the appropriate in-text citation.

“Although research is limited, some findings suggest that high FoMO individuals are more likely to engage in low-level illegal behavior such as driving while using a cell phone (Przybylski et al., 2013).”

178 I miss here a logic for titeling.

All titles have been fixed.

Part 1 [no subtitle] is about content with hypothesis for inference statistics

All titles have been fixed.

Part 2: "Statistical Modeling Approaches" is about machine learning techniques

All titles have been fixed.

207 evtl. missing reference "(2020)"

Thank you for pointing this out - we have fixed it.

“They further discussed the compatibility of machine learning alongside theoretical frameworks in psychological research (Elhai & Montag, 2020).”

Methods

279ff "While clinical diagnosis is slowly moving toward more dimensional approaches, diagnostic classification remains the long-established norm, especially in clinical practice (Woo & Keatinge, 2016)."

That's an argument. But your instruments are not constructed to make diagnoses. So, there is an annalogie to clinical diagnostics, but here it seems that you actually do clinical diagnostic classification.

It is meant as a justification for collapsing into binary categories for data analytic purposes. We explained how that was the best approach in this case. The reference to diagnostic classification was intended to contextualize this decision and explain that it is not out of the norm or uncalled for, as pertains to clinical practice. At no time did we render actual "diagnoses." That is, classifying a student as having engaged in a specific behavior in college is far, far from rendering a clinical diagnosis. We respectfully ask the review to consider this explanation and rationale.

307 Chapter number 2.3 already used on line 256

All titles have been fixed - thank you.

331 "review (Kotsiantis, 2007)" -> "review Kotsiantis (2007) for"

This has been updated accordingly.

Results

Table 1: The Cronbach's Alpha in line 1 and 2 of the table are confusing; they seem to be correlations with the variables itself. I would report these values in the text and not in this table.

This has been fixed so as to avoid confusing readers.

“Note. Chronbach’s Alpha for FoMO and Classroom Incivility are 0.894 and 0.856, respectively. Coefficients significant at p < .05 in bold. Coefficients significant at p < .01 in bold italics.”

after line 409 the numbering stops

This seems to be due to some error on the Submission Portal side. We will do our best to make sure that this same issue does not occur again.

Discussion

"General" I would change this subtitle to "Summary"

This has been changed as such.

General

I would use the word gender instead of the word sex.

Thank you for pointing this out. We have replaced “sex” with “gender” where appropriate in the paper. 

Reviewer #3: The paper is highly commendable. The topic is very timely and the analysis using the different data analytical tools produced impressive findings that could help scientist and experts in the field of behavioral sciences understand what FOMO is. However, the abstract should be improved. It only focused on the data analytical tools instead of the results and conclusions derived from the study which could help the readers understand what FOMO is and its relationship with some maladaptive behaviors. Authors may consider addressing this issue. Also, proper documentation of in-text citations should be observed. A large majority of the authors and works cited in the text are not listed in the references. This may derail the brevity of the study and its findings. All in all, the paper is highly acceptable.

We thank you for your time reviewing this as well as the suggestions to make this work even stronger.

---

## [Decision Letter · Decision Letter 1]

2 Sep 2022

College Student Fear of Missing Out (FoMO) and Maladaptive Behavior: Traditional Statistical Modeling and Predictive Analysis using Machine Learning

PONE-D-21-36817R1

Dear Dr. McKee,

We’re pleased to inform you that your manuscript has been judged scientifically suitable for publication and will be formally accepted for publication once it meets all outstanding technical requirements.

Kind regards,

Miquel Vall-llosera Camps

Senior Editor

PLOS ONE

Reviewers' comments:

Reviewer's Responses to Questions

**Comments to the Author**

1. If the authors have adequately addressed your comments raised in a previous round of review and you feel that this manuscript is now acceptable for publication, you may indicate that here to bypass the “Comments to the Author” section, enter your conflict of interest statement in the “Confidential to Editor” section, and submit your "Accept" recommendation.

Reviewer #1: All comments have been addressed

Reviewer #2: All comments have been addressed

2. Is the manuscript technically sound, and do the data support the conclusions?

Reviewer #1: Yes

Reviewer #2: Yes

3. Has the statistical analysis been performed appropriately and rigorously? 

Reviewer #1: Yes

Reviewer #2: Yes

4. Have the authors made all data underlying the findings in their manuscript fully available?

Reviewer #1: Yes

Reviewer #2: Yes

5. Is the manuscript presented in an intelligible fashion and written in standard English?

Reviewer #1: Yes

Reviewer #2: Yes

6. Review Comments to the Author

Reviewer #1: The authors did a thorough job addressing all of my concerns, clarifying, expanding, or editing as suggested. I believe this has made their strong work even stronger and recommend that the manuscript be accepted for publication.

Reviewer #2: I like the revised version of the paper very much. All the reviewers' questions and comments have been incorporated or answered satisfactorily.

I wish the authors that the paper will be cited frequently.

7. PLOS authors have the option to publish the peer review history of their article (what does this mean?). If published, this will include your full peer review and any attached files.

Reviewer #1: No

Reviewer #2: No

---

## [Editor Report · Acceptance letter]

9 Sep 2022

PONE-D-21-36817R1 

College Student Fear of Missing Out (FoMO) and Maladaptive Behavior: Traditional Statistical Modeling and Predictive Analysis using Machine Learning 

Dear Dr. McKee:

I'm pleased to inform you that your manuscript has been deemed suitable for publication in PLOS ONE. Congratulations! Your manuscript is now with our production department. 

Kind regards, 

on behalf of

Dr. Miquel Vall-llosera Camps 

Staff Editor

PLOS ONE